# Phosphoinositides regulate force-independent interactions between talin, vinculin, and actin

**Charlotte F Kelley[1]\*, Thomas Litschel[2], Stephanie Schumacher[1], Dirk Dedden[1], Petra Schwille[2], Naoko Mizuno[1,3,4]\***

[1]Max Planck Institute of Biochemistry, Department of Structural Cell Biology, Martinsried, Germany; [2]Max Planck Institute of Biochemistry, Department of Cellular and Molecular Biophysics, Martinsried, Germany; [3]Laboratory of Structural Cell Biology, National Heart, Lung, and Blood Institute, National Institutes of Health, Bethesda, United States; [4]National Institute of Arthritis and Musculoskeletal and Skin Diseases, National Institutes of Health, Bethesda, United States

**Abstract** Focal adhesions (FA) are large macromolecular assemblies which help transmit mechanical forces and regulatory signals between the extracellular matrix and an interacting cell. Two key proteins talin and vinculin connecting integrin to actomyosin networks in the cell. Both proteins bind to F-actin and each other, providing a foundation for network formation within FAs. However, the underlying mechanisms regulating their engagement remain unclear. Here, we report on the results of in vitro reconstitution of talin-vinculin-actin assemblies using synthetic membrane systems. We find that neither talin nor vinculin alone recruit actin filaments to the membrane. In contrast, phosphoinositide-rich membranes recruit and activate talin, and the membrane-bound talin then activates vinculin. Together, the two proteins then link actin to the membrane. Encapsulation of these components within vesicles reorganized actin into higher-order networks. Notably, these observations were made in the absence of applied force, whereby we infer that the initial assembly stage of FAs is force independent. Our findings demonstrate that the local membrane composition plays a key role in controlling the stepwise recruitment, activation, and engagement of proteins within FAs.

**\*For correspondence:**
kelley@biochem.mpg.de (CFK);
mizuno@biochem.mpg.de (NM)

**Competing interests:** The authors declare that no competing interests exist.

## Introduction

Focal adhesions (FAs) are highly dynamic protein complexes that direct cell migration, morphology, and differentiation. These assemblies require precise regulation of FA protein dynamics, often involving complex autoinhibitory mechanisms (*Khan and Goult, 2019*). The core structural FA proteins talin and vinculin connect integrin receptors at the plasma membrane to contractile actin networks within the cell; these interactions are critical for FA maturation, stability, and dynamics (*Atherton et al., 2015*; *Calderwood et al., 1999*; *Dumbauld et al., 2013*; *Goult et al., 2013b*; *Tadokoro et al., 2003*; *Thompson et al., 2014*). Both proteins are tightly autoregulated when not engaged within FAs, with many of their important binding sites occluded by intramolecular interactions (*Bakolitsa et al., 2004*; *Dedden et al., 2019*; *Goult et al., 2013b*; *Johnson and Craig, 1994*; *Johnson and Craig, 1995a*). How these proteins are recruited to, activated at, and disengaged from FAs is central to understanding FA regulation and cell migration processes.

FAs are organized into layers, each with distinct protein composition and function, spanning roughly 200 nm from the outside of the cell through the plasma membrane into the cytoplasm (*Kanchanawong et al., 2010*). The first of these, the integrin signaling layer, centers around activated integrin receptors, which directly connect cytoplasmic signaling and scaffolding proteins to

the extracellular matrix and to other membrane-interacting FA proteins. As integrin receptors have only short cytoplasmic domains, FAs require adaptor proteins to link the cytoskeleton to the membrane, making up the second layer of organization, the force transduction layer. These structural scaffolds transduce tension applied via contractile actomyosin networks across FA complexes, which is critical for growth and maturation (*Thompson et al., 2014*; *Wolfenson et al., 2011*). Mechanosensitive proteins bind directly to actin filaments in the actin regulatory layer, which also contains many actin-binding proteins that modulate actin dynamics and organization.

Talin is the only protein to extend from the base of FAs at the plasma membrane to the dense actin fiber network, interacting directly with both integrin receptors and the actin cytoskeleton (*Kanchanawong et al., 2010*). Talin is a 270 kDa protein, composed of an N-terminal globular head, a mechanosensitive rod composed of 13 helical bundles, which unfolds to an extended conformation up to 60–100 nm long when fully engaged, and a proposed dimerization domain at the C-terminus (*Calderwood et al., 2013*; *Goult et al., 2013a*; *Liu et al., 2015*). The talin head consists of a 4.1-ezrin-radixinmoesin (FERM) domain with four subdomains (F0-F3), which contain an integrin-binding site and a phosphoinositol-4,5-bisphosphate (PI(4,5)P$_2$) interaction surface (*Calderwood et al., 2013*). While the talin head binds to and activates integrin receptors, the talin rod domain contains critical actin-binding sites (*Atherton et al., 2015*; *Hemmings et al., 1996*). The structure of full-length human talin1 recently revealed that the inhibited protein takes on a compact globular conformation, maintained by numerous charge-based intramolecular interactions between the rod domains and secured by interactions with the FERM domain (*Dedden et al., 2019*). Although the autoinhibited structure offers insight into talin regulation, the question of what triggers the transformation of talin into an open, extended conformation remains unclear.

While talin provides a direct link between integrin receptors and the actin cytoskeleton, this connection is enhanced through interactions with vinculin (*Thievessen et al., 2013*; *Thompson et al., 2014*), another critical component of the mechanosensitive FA machinery (*Carisey et al., 2013*; *Diez et al., 2011*; *Dumbauld et al., 2013*). Vinculin interacts with both talin and actin (*Burridge and Mangeat, 1984*; *Jockusch and Isenberg, 1981*; *Wilkins and Lin, 1982*), and mediates force-dependent FA maturation, stability, and protein turnover (*Chen et al., 2005*; *Humphries et al., 2007*; *Saunders et al., 2006*). Although it has no known enzymatic activity, vinculin has at least 18 binding partners and acts as a key scaffold within adhesions (*Bays and DeMali, 2017*; *Carisey and Ballestrem, 2011*). Full-length vinculin autoinhibition is regulated by two distinct head-to-tail interactions, which result in an autoinhibitory interaction of nanomolar affinity (*Bakolitsa et al., 2004*; *Cohen et al., 2005*). Interrupting the vinculin head-to-tail interaction is predicted to require combinatorial activation via two or more ligands, as no single binding partner has been shown to overcome it (*Chen et al., 2006*).

In cells, talin mediates vinculin activation at FAs, while vinculin strengthens the connection between talin and the actin cytoskeleton (*Case et al., 2015*). Together, they provide the core mechanosensitive structure within FA complexes. Tension applied via actomyosin contraction can unfold the individual talin rod domains, revealing cryptic binding sites for vinculin (*Ciobanasu et al., 2014*; *Fillingham et al., 2005*; *Gingras et al., 2005*; *Papagrigoriou et al., 2004*; *del Rio et al., 2009*), enhancing connections to the actin network, and triggering maturation of FA complexes (*Atherton et al., 2015*; *Carisey et al., 2013*; *Humphries et al., 2007*; *Lee et al., 2013*; *Thompson et al., 2014*). However, how talin-vinculin-actin interactions are first initiated, and whether they can occur in the absence of applied force, have remained open questions. Investigations thus far have provided a limited view, as most in vitro studies have focused on interactions between individual domains or truncations, and have not investigated these questions in the context of the full-length proteins (*Cohen et al., 2006*; *Dedden et al., 2019*; *Gilmore and Burridge, 1996*; *Weekes et al., 1996*).

Vertebrates express two talin isoforms, encoded by *TLN1* and *TLN2*. (*McCann and Craig, 1997*) While they share 74% sequence identity, they are functionally distinct (*Debrand et al., 2012*; *Monkley et al., 2000*; *Monkley et al., 2001*). Talin1 is ubiquitously expressed and required during development, while talin2 is enriched in the brain and striated muscle, where its loss can be compensated for by talin1 (*Manso et al., 2017*; *Senetar et al., 2007*). Interestingly, talin2 often localizes to larger, more stable FAs, has a higher affinity for particular integrin receptors, and a greater specificity for alpha-actin, when compared to talin1 (*Franco et al., 2006*; *Senetar et al., 2004*; *Manso et al., 2013*; *Manso et al., 2017*; *Praekelt et al., 2012*; *Qi et al., 2016*). As our goal was to

investigate the underlying mechanisms regulating talin-vinculin-actin interactions using the simplest system possible, we focused on the talin2 isoform, allowing us to characterize a talin-vinculin-actin complex. Importantly, we accomplished this in the absence of applied force, indicating that while tension may be critical for downstream events related to FA assembly and maturation, initial talin-vinculin-actin interactions can be force independent.

Here, we characterize the interactions between full-length talin2, full-length vinculin, and actin in vitro. Using a variety of synthetic membrane systems, we have reconstituted talin-vinculin-mediated recruitment of actin to phospholipid bilayers, and established a robust system for further membrane-based reconstitution and analysis of minimal FA complexes. Importantly, these experiments elucidate mechanisms of activation for both talin and vinculin, lending much-needed insight into how assembly is initiated as well as the implications of their autoinhibitory mechanisms. Our results demonstrate that membrane binding facilitates activation of full-length talin2, which in turn recruits and activates full-length vinculin, thereby linking F-actin to PI(4,5)P$_2$-rich membranes in vitro.

## Results

### Autoinhibition blocks interactions between talin, vinculin, and actin in vitro

In order to isolate the regulatory mechanisms underlying talin-vinculin interactions in isolation, we purified the full-length proteins vinculin (Vn) and talin2 (Tn2) (*Figure 1A,B*) recombinantly. Consistent with previous findings (*Cohen et al., 2006*; *Dedden et al., 2019*), the wild-type proteins did not interact stably under either low or high ionic strength conditions during size-exclusion chromatography (*Figure 1—figure supplements 1* and *2*). We also tested the double mutant vinculin$^{N773A,E775A}$ (Vn$^{2A}$), as these mutations disrupt the interaction between vinculin D4 and tail domain (*Figure 1C*), thereby weakening the overall head-tail autoinhibitory interaction (*Cohen et al., 2005*). At low ionic strength, Vn$^{2A}$ and Tn2 also failed to form a detectable complex (*Figure 1—figure supplement 1*), but the two proteins co-migrated at higher ionic strength, indicating stable complex formation (*Figure 1—figure supplement 2*). These results are consistent with experiments carried out with Tn1, which assumes a compact, autoinhibited conformation at low ionic strength, but unfolds to ~60 nm in length when ionic strength is increased, revealing a vinculin-binding site (*Dedden et al., 2019*). Dynamic light-scattering (DLS) measurements indicate that Tn2 undergoes a similar transition (*Figure 1—figure supplement 3*). This indicates that autoinhibition of talin and vinculin each represent an independent barrier to complex formation, and that it is necessary for both to be released in order for talin and vinculin to stably interact.

Both talin and vinculin have actin-binding sites, which are implicated in their mechanosensitivity, their interactions with each other, and the organization of FAs (*Atherton et al., 2015*; *Thompson et al., 2014*). To test whether autoinhibition of talin or vinculin is relieved in the presence of actin, we used SNAP-tag-labeled versions of the full-length proteins to directly observe their interactions using total internal reflection fluorescence microscopy (TIRFm). While binding appeared weak for the individual proteins, enrichment was observed for both proteins along actin filaments when incubated together (*Figure 1D–F*, *Figure 1—source data 1*). We confirmed this effect using an actin co-sedimentation assay, which indicated an increase in actin co-sedimentation for both Tn2 and Vn$^{2A}$ when incubated together (*Figure 1—figure supplement 4*). Interestingly, Tn, Vn, and Vn$^{2A}$ preferentially bound to actin bundles of multiple filaments when incubated together (*Figure 1—figure supplement 3*), which we observed with increased frequency when actin was polymerized in the presence of both Tn and Vn$^{2A}$ (*Figure 1G*, *Figure 1—source data 1*). Overall, these results suggest that while actin is not sufficient to induce talin-vinculin interactions or release inhibition of either protein alone, it mediates talin-vinculin interactions if vinculin autoinhibition is partially relieved using Vn$^{2A}$.

### Deregulated vinculin induces actin bundling with full-length talin

Both talin purified from avian gizzard and the isolated vinculin tail domain have been shown to bundle F-actin in vitro (*Jockusch and Isenberg, 1981*; *Wilkins and Lin, 1982*; *Zhang et al., 1996*). Here, we used a low-speed co-sedimentation assay to detect actin bundling with the full-length recombinantly purified proteins. No bundling was detected for actin in the presence of either Tn2 or

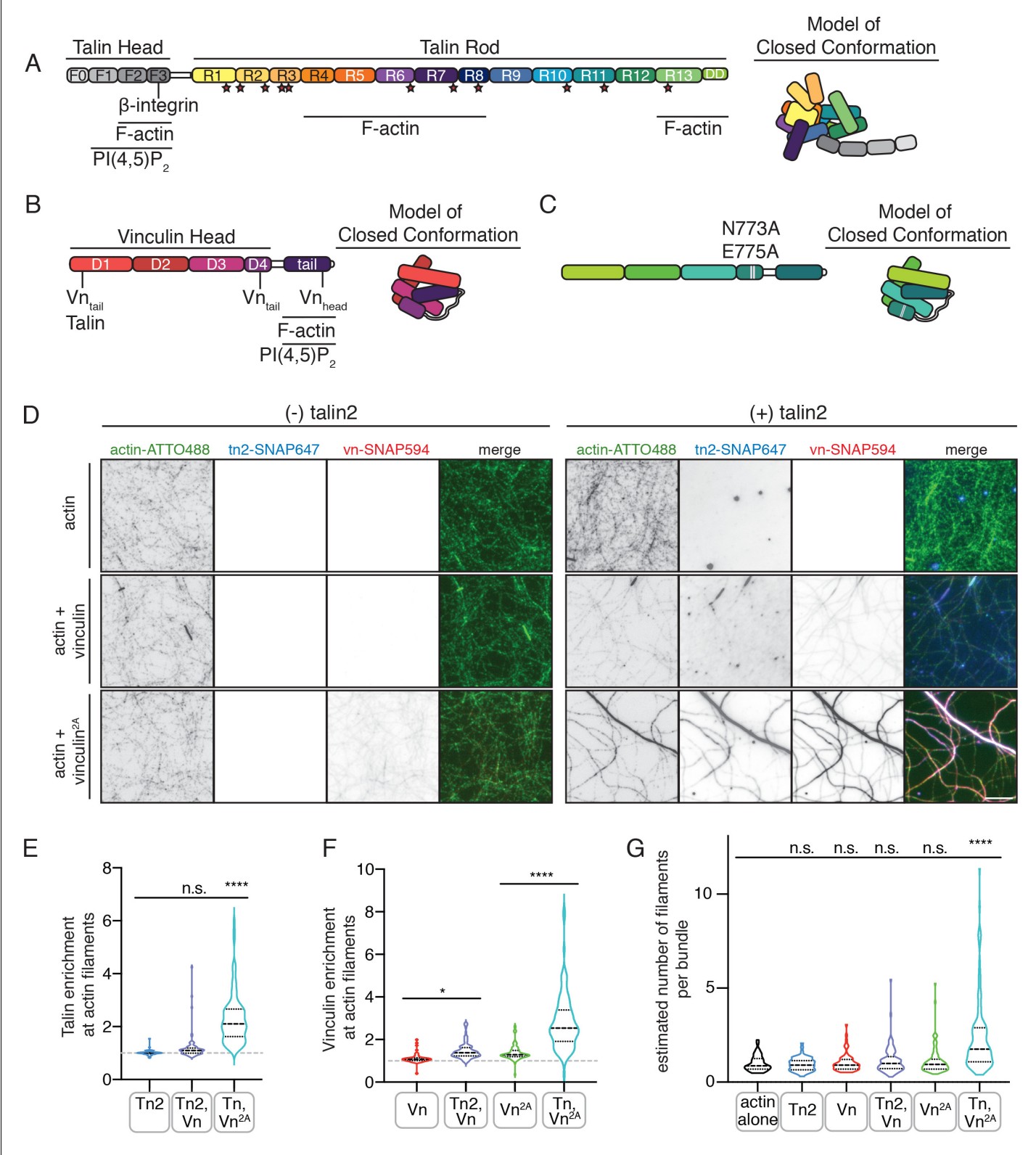

**Figure 1.** Autoinhibition blocks interactions between talin, vinculin, and actin in vitro. (**A**) Human talin2 domain organization, left. Stars highlight predicted vinculin binding sites. To the right, a model of the closed, autoinhibited conformation of talin, based on the Tn1 autoinhibited structure (***Dedden et al., 2019***). (**B**) Human vinculin domain organization, with model of autoinhibited structure (red) and (**C**) human vinculin with a deregulating mutation in the D4 domain (N773A,E775A), with a model of the partially deregulated structure (green) (***Bakolitsa et al., 2004***; ***Cohen et al., 2005***). (**D**)

*Figure 1 continued on next page*

*Figure 1 continued*

Representative three-color TIRF microscopy images of 1.5 µM SNAP-tag-labeled FA proteins and 1 µM actin (10% actin-ATTO488,green) added to TIRFm buffer (10 mM imidazole, 50 mM KCl, 1 mM MgCl$_2$, 1 mM EGTA, 0.2 mM ATP, pH 7.5) supplemented with 15 mM glucose, 20 µg/mL catalase, 100 µg/mL glucose oxidase, 1 mM DTT and 0.25% methyl-cellulose (4000 cp). Images acquired after 30 min of polymerization. Scale bar = 5 µm. (E) Quantification of talin enrichment at actin filaments, compared to background Tn-SNAP647 signal, for Tn alone, with Vn wild-type, and with Vn$^{2A}$. A value of 1 indicates no enrichment at filaments. (F) Quantification of vinculin enrichment at actin filaments, compared to background Vn-SNAP647 or Vn$^{2A}$-SNAP647 signal, both with or without Tn2. A value of 1 indicates no enrichment at filaments. (G) Estimation of number of filaments per actin bundle for each condition shown in (D), n (from left to right)=43, 57, 60, 93, 97, 108. The average fluorescence of individual filaments in actin alone samples was used to define the signal of a single actin filament, and then used to estimate the number of filaments for each condition. n.s. >0.5, *p<0.05 ****p<0.0001, by one-way ANOVA.

The online version of this article includes the following source data and figure supplement(s) for figure 1:

**Source data 1.** source data corresponding to *Figure 1D–G*.
**Figure supplement 1.** Talin and vinculin do not form a complex under low ionic strength conditions.
**Figure supplement 2.** Partial relief of vinculin inhibition allows talin-vinculin complex formation under high ionic strength conditions.
**Figure supplement 3.** Talin and vinculin are enriched along multi-filament actin bundles.
**Figure supplement 4.** Talin and vinculin interact weakly with actin independently.

Vn$^{2A}$ alone, but a large fraction of actin was present in the low-speed pellet fraction in the presence of Tn2 and Vn$^{2A}$ together (*Figure 2A*). The effect is consistent whether the proteins are incubated with pre-polymerized F-actin (*Figure 2A*) or with G-actin under polymerizing conditions (*Figure 2—figure supplement 1*). A titration of talin2 and vinculin indicates that maximal bundling is reached with a molar ratio of 1:1:1 talin2 to vinculin to actin (*Figure 2—figure supplement 2*). These results suggest that Tn2 and Vn$^{2A}$ work together to reorganize F-actin, while actin mediates the interaction between the two FA proteins.

Using TIRFm, we directly observed the effect of unlabeled Tn2 and Vn on the organization of F-actin. After 15 min of polymerization, the individual proteins, or a mixture of Tn2 and Vn, have little observable effect on F-actin organization, while drastic F-actin bundling occurred in the presence of both Tn2 and Vn$^{2A}$ (*Figure 2—video 1*, Figure 2 - figure supplement 3). This effect was concentration dependent, consistent with bulk bundling assays, and could be quantitatively assessed by measuring (1) the fraction of area covered by actin and (2) the approximate number of actin filaments per bundle (*Figure 2B*, *Figure 2—source data 1*). As bundling increased, the percentage of actin coverage decreased, while the number of filaments per bundle increased. Examples of Tn2-Vn$^{2A}$-mediated filament cross-linking and bundling can be observed during actin polymerization (*Figure 2C*, *Figure 2—videos 2* and *3*). These results suggest that in the presence of actin, the threshold for overcoming talin autoinhibition is lowered such that a talin-vinculin$^{2A}$-actin complex can form, while the binary interactions do not occur independently under the same conditions.

## Talin mediates interactions between full-length vinculin and actin

We next characterized which binding sites are required for actin remodeling using vinculin fragments and talin2 actin binding site (ABS) mutants (*Figure 3—figure supplement 1*). Using vinculin truncations, we tested whether the actin binding of vinculin is required for actin crosslinking. D1 of vinculin head (residues 1–258, Vn$^{D1}$) includes the talin-binding site (*Bakolitsa et al., 2004*; *Jones et al., 1989*), but does not result in actin bundling in the presence of Tn2. Interestingly, vinculin head (residues 1–823, Vn$^{Head}$) resulted in intermediate bundling (*Figure 3A*, *Figure 3—figure supplement 2*). This indicates that the combination of Vn$^{Head}$ and Tn2 can induce bundling, independent of the actin-binding vinculin tail. Vinculin tail (residues 878–1066, Vn$^{Tail}$) is able to mediate actin bundling alone (*Figure 3A*, *Figure 3—figure supplement 2*), consistent with previous studies (*Johnson and Craig, 1995a*; *Johnson and Craig, 2000*; *Shen et al., 2011*). These results suggest that both talin and vinculin have the potential to contribute to actin reorganization through their respective actin binding sites. As the isolated vinculin tail domain is sufficient to induce actin bundling, it is possible that talin could act as a vinculin activator, releasing the inhibited actin bundling activity of the vinculin tail domain. However, a talin vinculin binding site (VBS) peptide was insufficient to induce actin bundling in the presence of Vn$^{2A}$ (*Figure 3—figure supplement 2*), despite reportedly interrupting vinculin D1-tail autoinhibition in vitro (*Bass et al., 2002*; *Witt et al., 2004*).

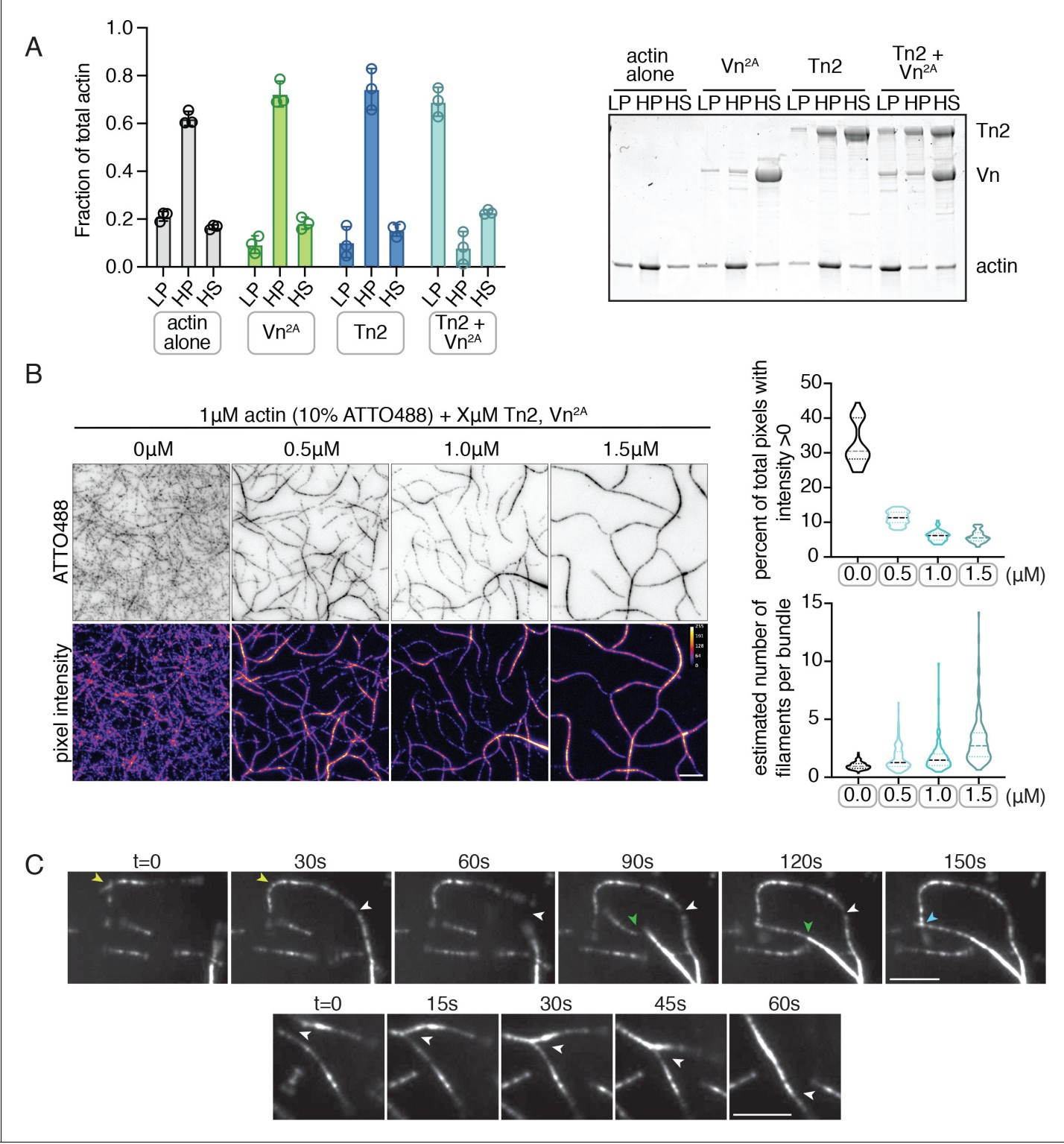

**Figure 2.** Deregulated vinculin induces actin bundling with full-length talin. (**A**) Actin bundling (low speed; 10,000 x g) and binding (high speed; 100,000 x g) co-sedimentation assay with pre-polymerized filamentous actin (2.5 µM) and purified FA proteins (2.5 µM). Low-speed pellet (LP), high-speed pellet (HP), and high-speed supernatant (HS) samples were analyzed by gradient SDS-PAGE. Graph shows individual data points, mean densitometry, and ± SD for three independent samples. See also *Figure 2—figure supplement 1* for controls and comparison to actin polymerized in the presence of talin and vinculin. (**B**) One-color TIRF microscopy of 1 µM actin with Tn2,Vn2A (0.5 to 1.5 µM) after 15 min of polymerization, also shown in *Figure 2—video 1*. Violin plots show percent area covered by actin for each condition tested (top) and the estimated number of filaments per bundle (bottom),
*Figure 2 continued on next page*

*Figure 2 continued*

based on the average peak fluorescence of individual filaments in the actin control sample, for three independent experiments. Percent area, from left to right, n = 31, 34, 45, 45. Filaments per bundle, from left to right, n = 117, 136, 184, 155. (C) Representative time-lapse images of 0.6 µM actin (5% ATTO488-actin) cross-linking and bundling during polymerization in the presence of 1 µM Tn2 and Vn$^{2A}$. Top images, multiple instances of filaments coming into contact and forming stable connections. Different colored arrows represent individual cross-linking events. Bottom images, filaments zipper together, white arrow. Scale bar = 5 µm. See also figure supplement videos 2 and 3. Cosedimentation experiments performed in 20 mM HEPES, pH 7.5, 75 mM KCl, 2 mM MgCl$_2$, and 0.2 mM ATP. TIRFm experiments carried out in TIRFm buffer with 15 mM glucose, 20 µg/mL catalase, 100 µg/mL glucose oxidase, 1 mM DTT and 0.25% methyl-cellulose (4000 cp).

The online version of this article includes the following video, source data, and figure supplement(s) for figure 2:

**Source data 1.** source data corresponding to *Figure 2B*.
**Figure supplement 1.** Actin also crosslinked when polymerized in the presence of talin and partially deregulated vinculin.
**Figure supplement 2.** Concentration dependence of talin-vinculin-mediated actin cross-linking.
**Figure supplement 3.** Both talin and vinculin are required for actin bundling.
**Figure 2—video 1.** Actin polymerization with Tn2, Vn or Tn2,Vn$^{2A}$.
https://elifesciences.org/articles/56110#fig2video1
**Figure 2—video 2.** Actin bundling by Tn2 and Vn$^{2A}$.
https://elifesciences.org/articles/56110#fig2video2
**Figure 2—video 3.** Actin bundling by Tn2 and Vn$^{2A}$.
https://elifesciences.org/articles/56110#fig2video3

Talin has three distinct ABSs: one located in the FERM domain (ABS1), one within rod domains 4 and 8 (ABS2), and one in rod domain 13 (ABS3) (*Figure 1A*; *Atherton et al., 2015*; *Gingras et al., 2008*; *Hemmings et al., 1996*; *McCann and Craig, 1997*). Tn$^{\Delta Head}$, lacking ABS1, was able to induce bundling similar to wild-type Tn2 with Vn$^{2A}$ (*Figure 3B*, *Figure 3—figure supplement 3*). To test the role of ABS2 and ABS3, the individual ABSs were mutated to reduce actin binding (Tn$^{ABS2}$, Tn2$^{ABS3}$, Tn2$^{ABS2/3}$ - *Atherton et al., 2015*; *Figure 3—figure supplement 1*). Similar to Tn2, these mutants were unable to bundle actin alone, but had a strong effect in the presence of Vn$^{2A}$ (*Figure 3B*). To confirm that neither of the talin2 ABSs within the rod domain was required for this effect, a fragment truncated after R3 was tested (residues 1–910, Tn$^{N:R3}$). Again, this fragment was not significantly different from Tn2 alone or in the presence of Vn$^{2A}$ (*Figure 3B*).

To identify any differences in binding or bundling that would be missed in a bulk co-sedimentation assay, SNAP-tagged versions of Tn2$^{ABS2}$, Tn2$^{ABS3}$, Tn2$^{ABS2/3}$, and Tn2$^{N:R3}$ were directly observed using TIRFm (*Figure 3C*). In the presence of Vn$^{2A}$, all associated with actin filaments and induced bundling. None of the actin binding-deficient Tn2 constructs were significantly different from wild-type in terms of the number of filaments per bundle (*Figure 3—figure supplement 4*, *Figure 3—figure supplement 4—source data 1*), although some slight differences in actin binding were detected. While Tn2 does not strongly associate with single filaments, Tn2$^{ABS2}$ showed significantly more enrichment along individual or pairs of filaments, as did Tn2$^{ABS2/3}$ and Tn2$^{N:R3}$, though to a lesser extent. There were no significant differences between Tn2 and Tn2$^{ABS3}$ for either single, paired, or bundled actin filaments (*Figure 3—figure supplement 4*, *Figure 3—figure supplement 4—source data 1*).

Overall, these results suggest that no single ABS is absolutely required for talin-vinculin mediated actin crosslinking. Indeed, mutation of ABS2 increases association with actin filaments, and decreases specificity for bundled actin filaments. It is possible that mutations in ABS2 may affect the strength of interactions between talin rod domains required for the autoinhibited conformation, weakening autoregulatory interactions, lowering the threshold for vinculin binding, and increasing the recruitment of talin and vinculin to actin filaments. In support of this hypothesis, more vinculin co-sediments with actin bundles in the presence of Tn2$^{ABS2}$ and Tn2$^{ABS2/3}$ (*Figure 3—figure supplement 3*). As this effect is also present in the double ABS2/ABS3 mutant, these results strongly suggest that the majority of actin bundling is mediated by direct interactions between actin and the vinculin tail domain, which only occur when the full-length vinculin is in the presence of talin domains.

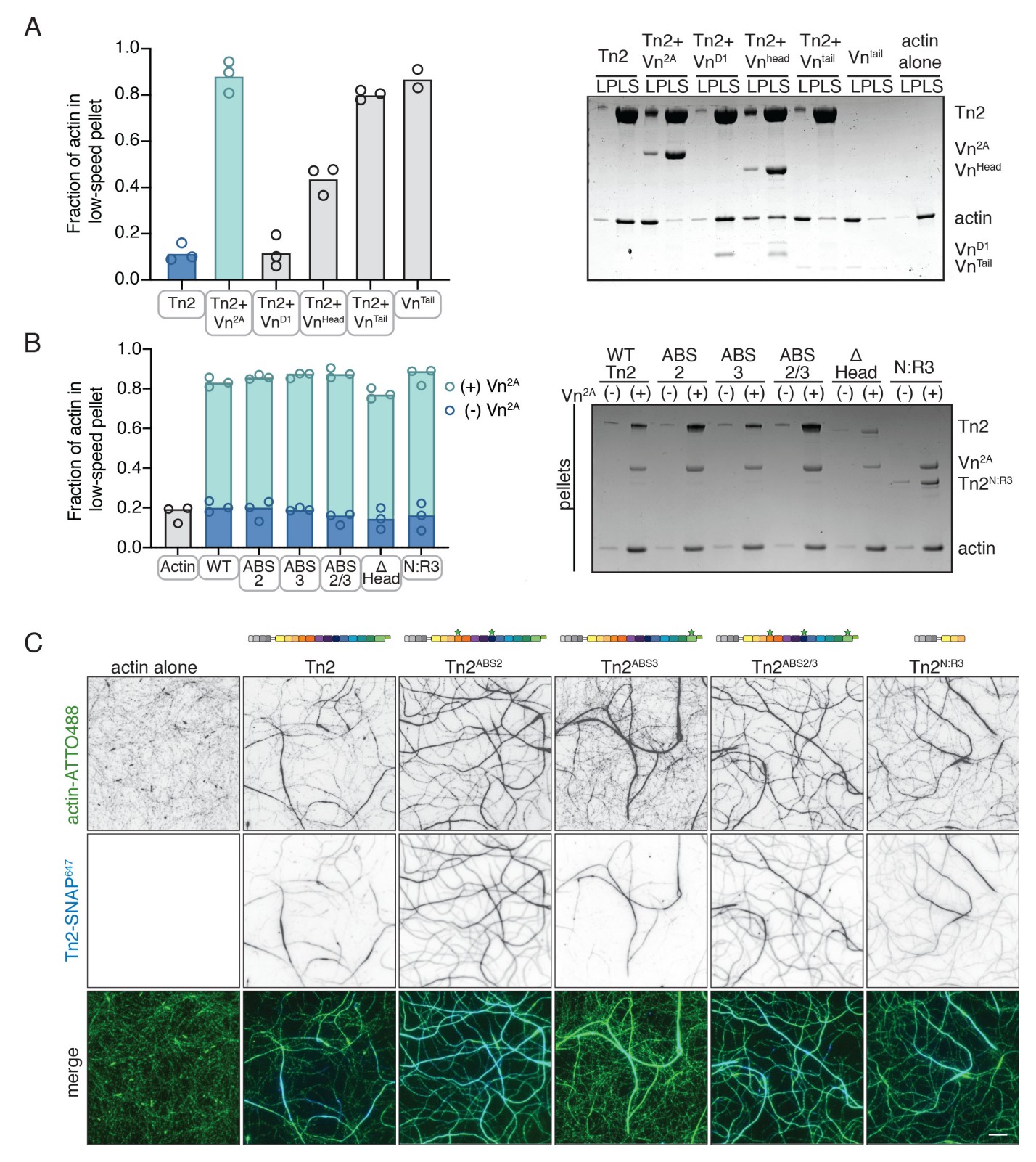

**Figure 3.** Talin mediates interactions between full-length vinculin and actin. (A) Actin bundling co-sedimentation assay with filamentous actin (2.5 μM) and Tn2 with various vinculin fragments (2.5 μM, see *Figure 3—figure supplement 1*), supernatant and pellet samples were analyzed using SDS-PAGE. Graph shows individual data points and mean densitometry for three independent samples. (B) Actin bundling co-sedimentation assay with filamentous actin (2.5 μM) and various talin2 mutants and fragments (see *Figure 3—figure supplement 1*), with and without $Vn^{2A}$ (2.5 μM), supernatant

*Figure 3 continued on next page*

*Figure 3 continued*

and pellet samples were analyzed using SDS-PAGE. Graph shows individual data points and mean densitometry for talin2 proteins alone (dark blue) and with $Vn^{2A}$ (aqua) for three independent samples. Cosedimentation experiments performed in 20 mM HEPES, pH 7.5, 75 mM KCl, 2 mM $MgCl_2$, and 0.2 mM ATP. Controls for (**A,B**) found in *Figure 3—figure supplement 2*.(**C**) Representative images of two-color TIRFm with 1 µM actin (5% ATTO488-actin) and $Vn^{2A}$ (1.5 µM) with different SNAP-647-labeled Tn2 proteins (1.5 µM). Schematics above images indicate domain locations of mutations and truncations. Quantification can be found in *Figure 3—figure supplement 4*. Scale bar = 5 µm. TIRFm experiments were carried out in TIRFm buffer with 15 mM glucose, 20 µg/mL catalase, 100 µg/mL glucose oxidase, 1 mM DTT and 0.25% methyl-cellulose (4000 cp).

The online version of this article includes the following source data and figure supplement(s) for figure 3:

**Figure supplement 1.** Domain schematics of talin mutants and vinculin truncations.
**Figure supplement 2.** Cosedimentation controls corresponding to *Figure 3*.
**Figure supplement 3.** Mass spectrometry of $Tn2^{\Delta Head}$ and quantification of Vn cosedimentation with Tn2 mutants.
**Figure supplement 4.** Quantification of Tn2 mutant TIRFm data.
**Figure supplement 4—source data 1.** source data corresponding to *Figure 3—figure supplement 4*.

## $PI(4,5)P_2$-rich membranes unlock talin-vinculin interactions in vitro

Talin, actin, and phospholipid binding have all been implicated in vinculin regulation, but do not individually activate wild-type-vinculin. Thus, it is likely that vinculin requires combinatorial regulatory inputs to relieve autoinhibition. One likely candidate is $PI(4,5)P_2$, an important phospholipid present within the plasma membrane that plays key regulatory roles in actin dynamics (*Janmey et al., 2018*; *Zhang et al., 2012*). When $PI(4,5)P_2$ levels are reduced, cells fail to form new FAs or stress fibers (*Gilmore and Burridge, 1996*) and the recruitment of talin and vinculin to new adhesions is impaired (*Legate et al., 2011*). The talin FERM domain and vinculin tail domain both interact with $PI(4,5)P_2$ in vitro, while $PI(4,5)P_2$-binding deficient mutants of the full-length proteins have been reported to disrupt actin cytoskeleton organization, cell spreading, and migration (*Chinthalapudi et al., 2014*; *Chinthalapudi et al., 2018*). In addition, $PI(4,5)P_2$ has previously been reported to disrupt vinculin head-tail interactions (*Johnson and Craig, 1995a*, *Gilmore and Burridge, 1996*; *Witt et al., 2004*). Therefore, we hypothesized that $PI(4,5)P_2$-containing membranes play a critical role in mediating engagement of FA proteins with the actin cytoskeleton.

In a liposome co-sedimentation assay, Tn2 interacted with synthetic liposomes in a $PI(4,5)P_2$-dependent manner, while neither Vn nor $Vn^{2A}$ co-sedimented with liposomes even at high levels of $PI(4,5)P_2$ (*Figure 4A*). This suggests that the D1-tail inhibitory interaction is sufficient for blocking interactions between the vinculin tail and membrane-embedded $PI(4,5)P_2$. Interestingly, in the presence of Tn2, both Vn and $Vn^{2A}$ are recruited to $PI(4,5)P_2$-rich membranes (*Figure 4A*). This effect requires the talin FERM domain, as neither $Tn2^{\Delta Head}$ nor the VBS peptide are able to induce $Vn^{2A}$ membrane-binding (*Figure 4—figure supplement 1*). Likewise, the presence of Vn or $Vn^{2A}$ increased the amount of membrane-bound Tn2 (*Figure 4A*).

To directly visualize talin2-mediated recruitment of vinculin, we incubated the SNAP-tagged proteins together with giant unilamellar vesicles (GUVs) containing 10% $PI(4,5)P_2$. In agreement with the co-sedimentation assay, Tn2-SNAP bound to the vesicle surface. Neither Vn nor $Vn^{2A}$ was recruited to the membrane surface in the absence of Tn2, while both were recruited in the presence of Tn2 (*Figure 4B*), and in turn increased the Tn2 signal. Furthermore, Vn intensity correlated with Tn2 intensity at the vesicle surface (*Figure 4—figure supplement 2*), suggesting their direct interactions at the membrane. Enrichment at the membrane was concentration-dependent, and peaks at 1 µM both for Tn2-Vn and Tn2-Vn2A (*Figure 4—figure supplement 2*). Above 1 µM, the membrane is likely saturated with protein, resulting in excess Tn2 and Vn in solution and a decrease in the measured enrichment at the GUV surface.

## Vinculin membrane binding is linked to autoinhibition

Talin binds to $PI(4,5)P_2$ via a binding site within its FERM domain (*Chinthalapudi et al., 2018*; *Song et al., 2012*), while vinculin contains two distinct lipid binding surfaces within the tail domain (*Chinthalapudi et al., 2014*; *Thompson et al., 2017*; *Figure 1A,B*). To identify the importance of each lipid binding site, we tested vinculin fragments for membrane co-sedimentation in the presence and absence of Tn2. Vn, $Vn^{D1}$, and $Vn^{Head}$ only co-sedimented with liposomes in the presence of Tn2 (*Figure 5—figure supplement 1*). $Vn^{Tail-SNAP}$ co-sedimented with $PI(4,5)P_2$-rich vesicles alone,

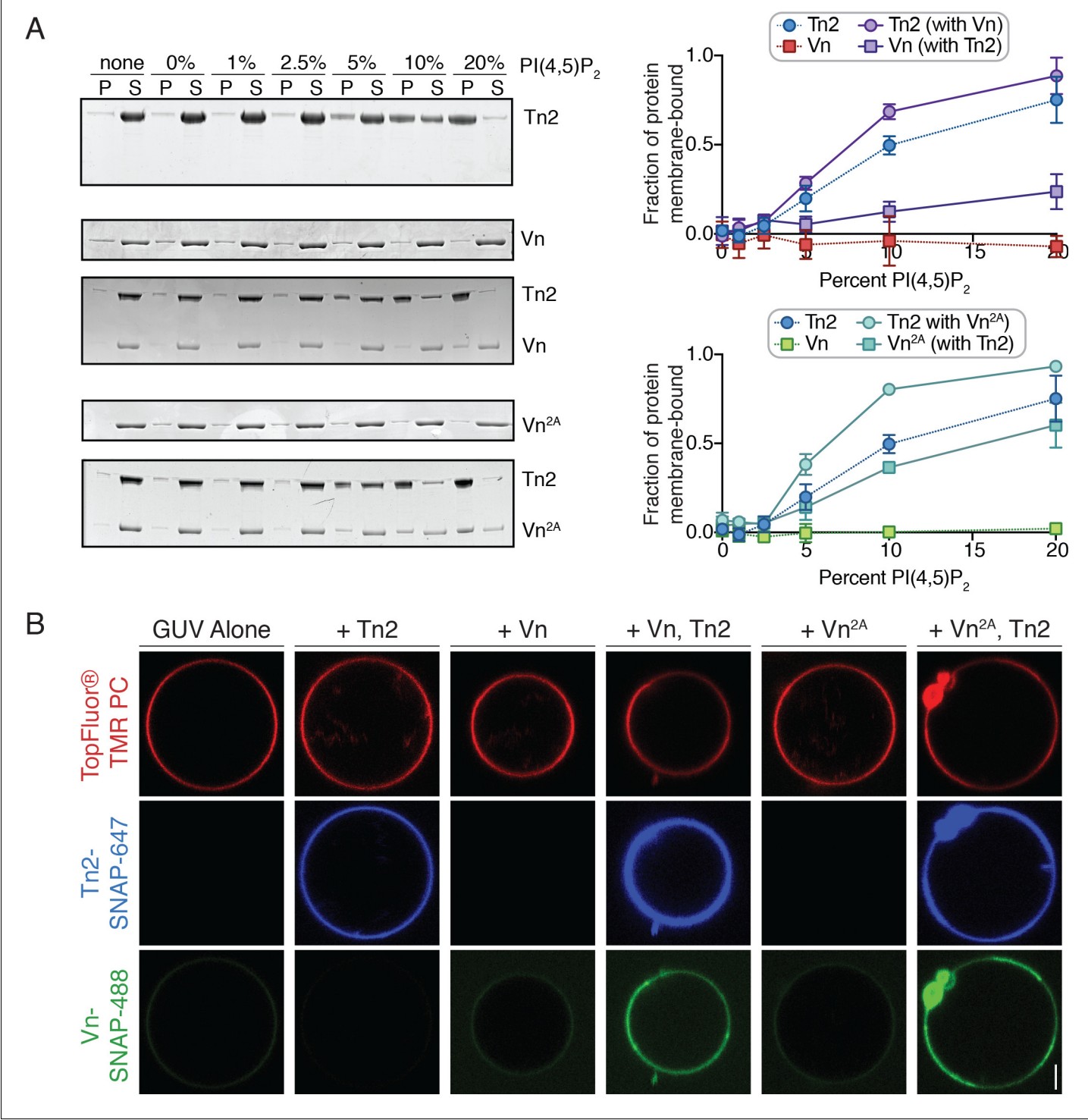

**Figure 4.** PI(4,5)P$_2$-rich membranes unlock talin-vinculin interactions in vitro. (**A**) Liposome co-sedimentation assay with purified Tn2, Vn, and Vn$^{2A}$ (0.5 μM). Proteins were incubated with liposomes of the following composition: 0.75-X DOPC, 0.15 DOPE, 0.1 DOPS, X PI(4,5)P$_2$. Representative Coomassie-stained SDS-PAGE of pellet and supernatant samples for each condition. Graphs show mean densitometry from three independent samples ± SD, for Tn2, Vn and Tn2, Vn$^{2A}$. Data for Tn2 alone (blue) is shown on both graphs, but represent identical data. (**B**) Representative images of PC-TMR-labeled GUVs (red) with purified, SNAP-tag-labeled FA proteins (Tn2: blue and Vn: green). For quantification see *Figure 4—figure supplement 2*. All liposome experiments in this figure were performed in liposome buffer (20 mM HEPES, pH 7.5 and 100 mM NaCl). Scale bar = 2.5 μm.

The online version of this article includes the following source data and figure supplement(s) for figure 4:

**Figure supplement 1.** Talin FERM domain is required to recruit vinculin to liposomes.

*Figure 4 continued on next page*

*Figure 4 continued*
**Figure supplement 2.** Quantification of talin-mediated vinculin recruitment to GUVs.
**Figure supplement 2—source data 1.** source data corresponding to *Figure 4—figure supplement 2*.

independent of Tn2 (*Figure 5B*, *Figure 1—figure supplement 1*). To further test whether vinculin is directly or indirectly recruited to lipid membranes in the presence of Tn2, we tested the triple mutant VnK944Q,R945Q,K1061Q (Vn$^{3Q}$) (*Figure 5—figure supplement 2*), reported to reduce vinculin-membrane interactions (*Chinthalapudi et al., 2014*; *Thompson et al., 2017*). Vn$^{3Q}$ disrupts both the vinculin tail basic collar (K1061; specific PI(4,5)P$_2$ interaction) and the basic ladder (K944, R945; anionic lipid-dependent insertion), which have both been implicated in membrane binding. In the isolated tail domain, the three mutations (Vn$^{Tail3Q}$) strongly decreased co-sedimentation with liposomes compared to the wild-type Vn$^{Tail}$ (*Figure 5—figure supplement 1*). Interestingly, the full-length Vn$^{3Q}$ co-sedimented significantly more than Vn in the presence of Tn2 (*Figure 5A*).

The recruitment of Vn$^{3Q}$ to membranes in this assay closely resembled that of Vn$^{2A}$, suggesting that the K944Q,R945Q,K1061Q mutations may have a secondary effect of disrupting autoinhibitory interactions within full-length vinculin. Upon closer inspection, the residues K944 and R945 are located at the vinculin tail-D1 interface, which forms the primary autoinhibitory interaction between vinculin head and tail domains (*Figure 5—figure supplement 2*; *Cohen et al., 2005*). To determine whether the 3Q lipid-binding-deficient mutation also partially relieves autoinhibition, we tested its interaction with Tn2 using size-exclusion chromatography. Surprisingly, Vn$^{3Q}$ co-migrated with Tn2 even under low ionic strength conditions (*Figure 5B*), unlike Vn$^{2A}$ (*Figure 1—figure supplement 2*). Vn$^{3Q}$ was indistinguishable from Vn$^{2A}$ in a bulk actin bundling pelleting assay, and localized to actin bundles with Tn2 in TIRFm experiments (*Figure 5C,D* and *Figure 5—figure supplement 1*). Overall, this strongly suggests that the mutant is partially deregulated, likely via disruption of the vinculin D1-tail interaction, and that talin can still recruit vinculin to membranes in the absence of vinculin lipid binding. Taken together, these results demonstrate that (i) proposed vinculin-membrane-binding sites are closely linked to autoregulation of the full-length protein and (ii) talin can recruit vinculin to PI(4,5)P$_2$-rich membranes independent of direct vinculin-membrane interactions.

## Membrane-bound talin and vinculin recruit actin to PI(4,5)P$_2$-rich membranes

The observation that wild-type vinculin is recruited to PI(4,5)P$_2$-rich membranes via a talin-dependent mechanism raises the question of whether membrane recruitment can activate talin-vinculin-actin interactions. To test this, an actin bundling co-sedimentation assay was performed with liposomes containing increasing amounts of PI(4,5)P$_2$. Interestingly, even at 5% PI(4,5)P$_2$, a condition at which talin2 and vinculin membrane binding is low, a large fraction of F-actin is in the low-speed pellet (*Figure 6A*). Tn2 was also able to induce actin co-sedimentation in the presence of 10% PI(4,5)P$_2$ liposomes, but the effect in the presence of vinculin was stronger. PI(4,5)P$_2$ liposomes also increased Tn2-Vn$^{2A}$ and Tn2-Vn$^{3Q}$-induced actin pelleting (*Figure 6—figure supplement 1*). Consistent with the recruitment of vinculin to PI(4,5)P$_2$-rich membranes and the actin bundling results, enhanced actin pelleting requires the talin FERM domain, but not talin ABS2 or ABS3 (*Figure 6—figure supplement 1*, *Figures 3B* and *5B*).

In order to directly observe actin recruitment to a phospholipid bilayer, we reconstituted talin2-vinculin-actin interactions on supported lipid bilayers (SLBs) containing 5% PI(4,5)P$_2$. Tn2 bound readily to SLBs, and recovered after photobleaching, although at a slower rate than lipids in the absence of protein (*Figure 6—figure supplement 2*, *Figure 6—figure supplement 2—source data 1*). The recovery of protein could indicate either exchange between membrane-bound protein and protein in solution, or lateral diffusion on the membrane (*Senju et al., 2017*). We also observed that lipid recovery is slightly slower in the presence of talin, indicating that talin can hinder lateral diffusion of lipids in the bilayer.

We preincubated Tn2 with the membrane for 15 min, added actin and vinculin (*Figure 6—figure supplement 3*), and allowed actin to polymerize for 45 min (*Figure 6—video 1*). In the presence of only Tn2, we observed very little actin accumulation on the membrane (*Figure 6B–D* and *Figure 6—video 1*). In the presence of all three proteins, vinculin was recruited to the SLB and actin became

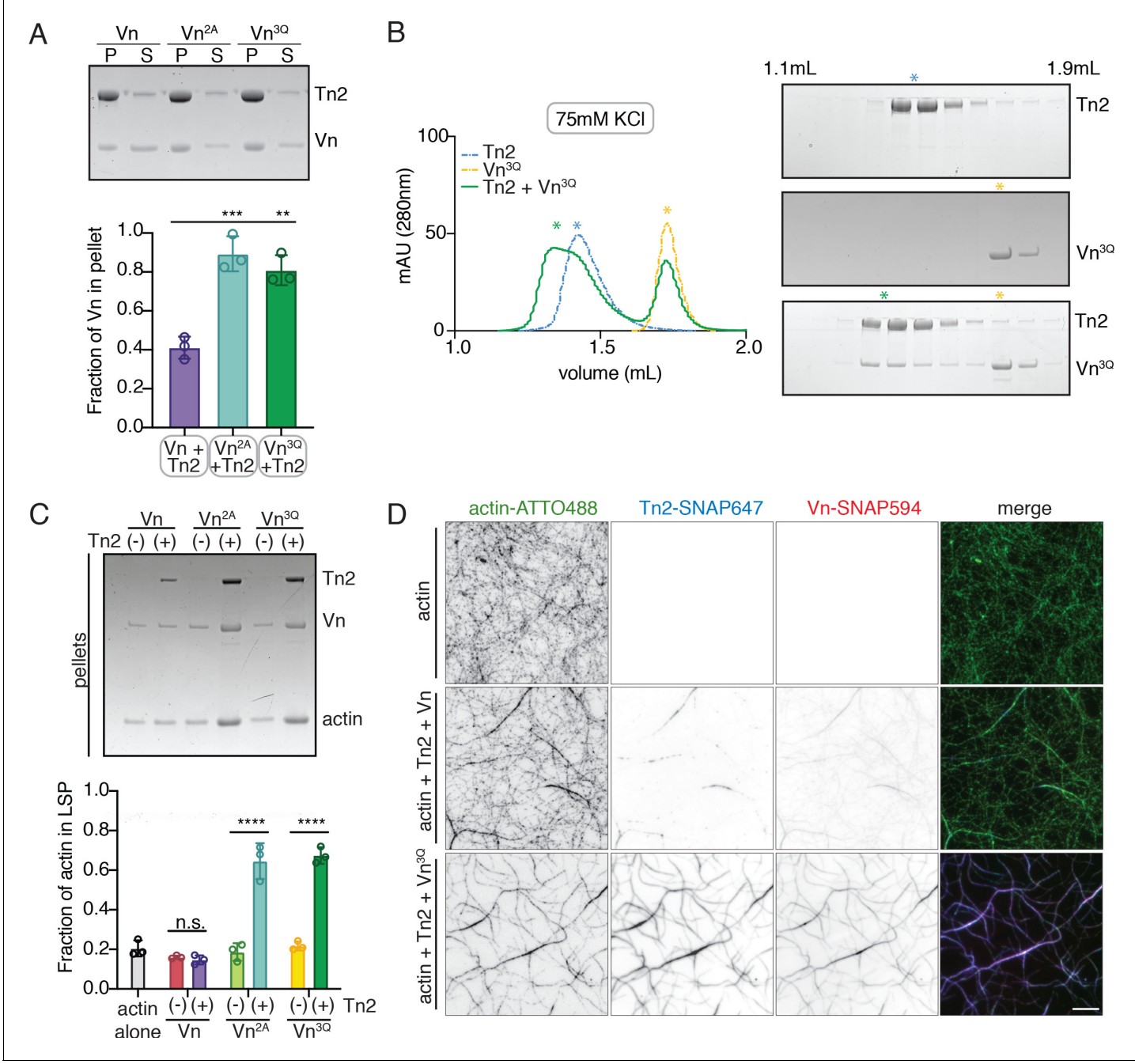

**Figure 5.** Vinculin membrane binding is linked to autoinhibition. (A) Liposome co-sedimentation assay with full-length Vn, Vn²ᴬ, Vn³ᵠ with Tn2 (0.5 µM) and liposomes (0.55 DOPC, 0.15 DOPE, 0.1 DOPS, 0.20 PI(4,5)P₂, molar ratio). Representative SDS-PAGE of pellet and supernatant samples, control found in *Figure 5—figure supplement 2*. Graph shows mean densitometry and individual values from three independent samples. All liposome cosedimentation assays in this figure were performed in liposome buffer. (B) Tn2 and Vn³ᵠ reconstitution assay using size-exclusion chromatography (SEC) in 20 mM HEPES pH 7.8, 75 mM KCl, 1 mM EDTA, 3 mM β-mercaptoethanol. Elution profiles compared by SDS-PAGE are on the right for Tn2 alone, Vn³ᵠ , and Tn2 with Vn³ᵠ (every other fraction loaded from 1.1 mL to 1.9 mL). (C) Actin bundling co-sedimentation assay with vinculin mutants, with and without Tn2, carried out under the following buffer conditions: 20 mM HEPES, pH 7.5, 75 mM KCl, 2 mM MgCl₂, and 0.2 mM ATP. Representative SDS-PAGE of pellet samples after low-speed spin (10,000 x g), control found in *Figure 5—figure supplement 1*. Graph represents individual and mean densitometry from three individual experiments. ****p<0.0005, one-way ANOVA. (D) Representative images of three-color TIRFm with Vn-SNAP594, Vn³ᵠ-SNAP594, and Tn2-SNAP647, carried out in TIRFm buffer with 15 mM glucose, 20 µg/mL catalase, 100 µg/mL glucose oxidase, 1 mM DTT and 0.25% methyl-cellulose (4000 cp). Conditions were each repeated in triplicate with consistent results. Scale bar = 5 µm.

The online version of this article includes the following figure supplement(s) for figure 5:

**Figure supplement 1.** Additional cosedimentation experiments and controls corresponding to *Figure 5*.

*Figure 5 continued on next page*

*Figure 5 continued*

**Figure supplement 2.** Lipid-binding residues are located at the vinculin tail-D1 interface.

enriched at the membrane surface as it polymerized (*Figure 6B–D*, *Figure 6—figure supplement 3*, and *Figure 6—video 1*). Protein signal in relation to the SLB was analyzed for three separate experiments. By using the z-position of peak talin signal as a proxy for the location of the membrane (z-position = 0), we were able to compare the recruitment of actin to the membrane in the presence or absence of vinculin. In the presence of Vn or Vn$^{2A}$, the peak actin signal consistently occurred at or very close to the membrane surface, on average under 500 nm for both conditions. In the absence of Vn or Vn$^{2A}$, the position of the peak actin signal varied greatly, and was, on average, greater than 2 μm (*Figure 6—figure supplement 3*). In addition, the raw actin signal at the membrane is significantly lower for Tn2 alone compared to in the presence of either Vn or Vn$^{2A}$ (*Figure 6C*, *Figure 6— figure supplement 3—source data 1*). This measurement confirmed that vinculin is required for significant enrichment of actin at the lipid bilayer. Interestingly, in the presence of Tn2 and Vn2A, the enrichment of actin at the membrane surface was not as specific, and actin bundles were also observed above the membrane surface. This can be seen in 3D reconstructions of confocal z-stacks for the three different conditions (*Figure 6—video 2*). Quantification of individual protein signals also reflected these observations. In the presence of Tn and Vn$^{2A}$, the signal above the membrane (z-position >0) is higher for all three proteins (Tn2, Vn$^{2A}$, actin) when compared to in the presence of Tn and Vn (*Figure 6—figure supplement 3B*, *Figure 6—figure supplement 2—source data 1*). This suggests that vinculin autoregulation plays a role in specifically targeting talin-vinculin-interactions to the membrane surface, whereas in the presence of deregulated Vn$^{2A}$, interactions between the three proteins are not wholly dependent on membrane binding. Thus, vinculin autoinhibition is important for limiting talin and vinculin interactions with actin to the membrane surface, preventing unspecific interactions along actin filaments further away from the plasma membrane.

## Encapsulated talin and vinculin reorganizes F-actin into highly bundled and cross-linked structures

The SLB experiments indicated that interactions with phospholipid bilayers play an important role in regulating talin and vinculin interactions with actin. In order to better understand the regulatory effect of a free-standing PI(4,5)P$_2$-containing bilayer, we encapsulated Tn2, Vn, and actin in vesicles. We mixed the proteins under polymerizing conditions immediately before encapsulation into GUVs using the continuous droplet interface crossing encapsulation (cDICE) method (*Figure 7A*; *Litschel et al., 2020*). This allows for better visualization of higher-order actin structures, and mimics compartmentalization within a cell-sized membrane compartment, resulting in individual, confined reactions on a much smaller scale than SLB experiments. We used the MinDE system, which is known to oscillate between bound and unbound to negatively charged membranes (*Litschel et al., 2018*), to confirm that PI(4,5)P$_2$ was successfully incorporated into the vesicles (*Figure 7—figure supplement 1*).

We did not observe any higher-order actin structures when actin was encapsulated alone, or in the presence of any of the individual FA proteins, while both Tn2 with Vn and Tn with Vn2A induce visible actin bundling (*Figure 7B,C*, *Figure 7—video 1*). In both cases, the talin and vinculin signal appeared along the bundled actin filaments. In order to compare the actin re-organizing activity of Tn with Vn or Vn$^{2A}$, we sorted GUVs into three classes, according to the degree of actin organization: (i) none (no visible actin bundling), (ii) low (visible higher-order actin structures, but undefinable number of bundles), and (iii) high (clear number of bundles) (*Figure 7D*). While both Tn2-Vn and Tn2-Vn$^{2A}$-containing GUVs show varying degrees of actin organization, a qualitative assessment of the observed actin structures indicated that the bundling activity of Tn2 with Vn$^{2A}$ is more extensive than that of Tn2 with Vn. A majority of vesicles encapsulating Tn2, Vn$^{2A}$, and actin contain a small number of thick actin bundles, often around the circumference of the vesicle. Notably, Tn2, Vn-containing vesicles are less uniform, and overall exhibit a lower degree of actin organization (*Figure 7D*), indicating higher activity of Tn2 with Vn$^{2A}$, consistent with our other experiments.

In both cases, any bundles present are coated with talin and vinculin, and often have at least one contact point with the membrane. Vesicles with a high degree of actin organization (*Figure 7D*)

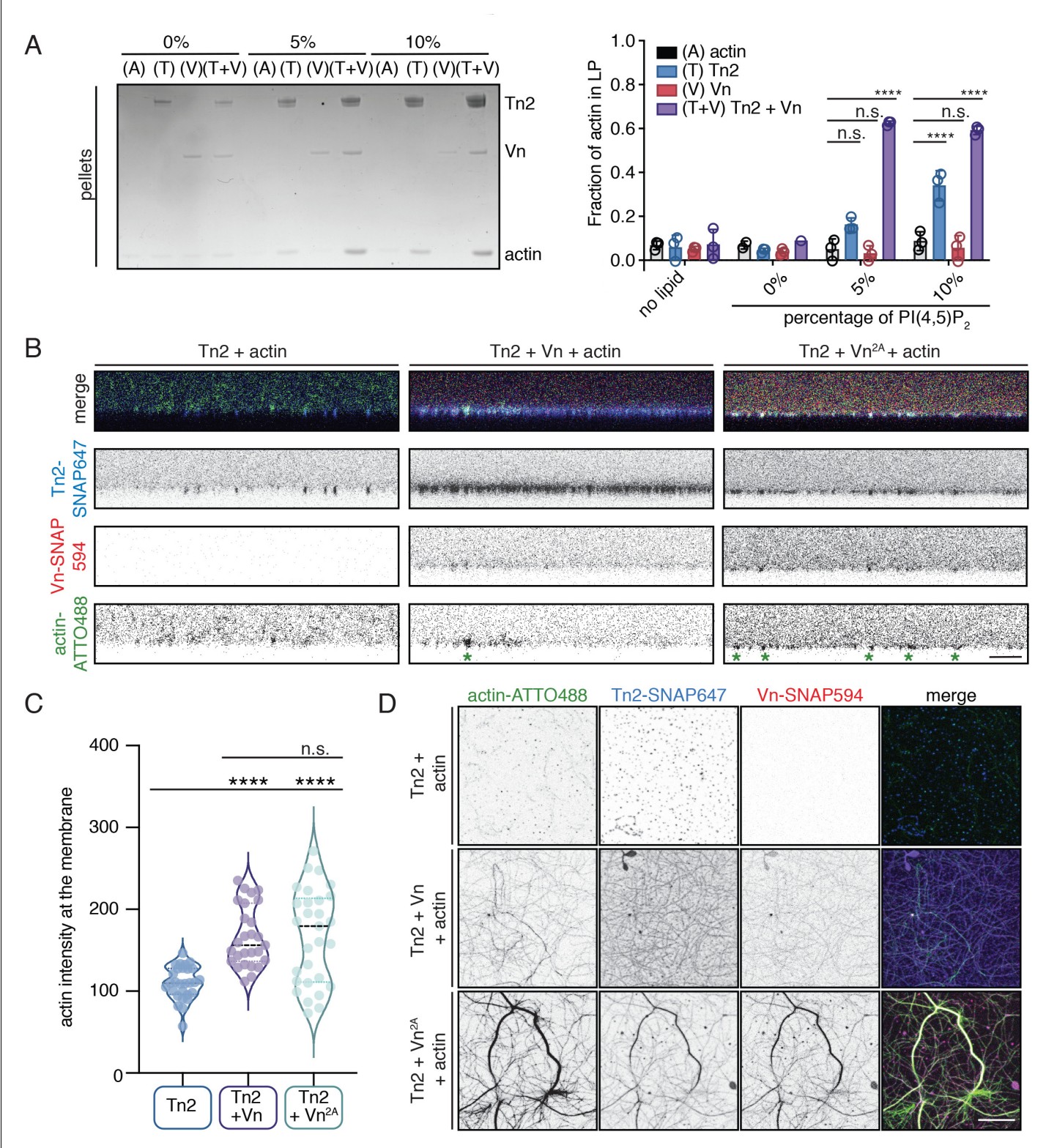

**Figure 6.** Talin and vinculin recruit actin to PI(4,5)P₂-rich membranes. (**A**) Actin bundling co-sedimentation assay with liposomes of the following composition: 0.75-X DOPC, 0.15 DOPE, 0.1 DOPS, X PI(4,5)P₂. FA proteins (2.5 μM) were pre-incubated with liposomes, followed by the addition of G-actin (2.5 μM). Final buffer conditions were 20 mM HEPES pH 7.5, 20 mM NaCl, 85 mM KCl, 1.6 mM MgCl₂, 0.16 mM ATP. After actin polymerization, samples were centrifuged to sediment liposomes and membrane-associated proteins (10,000 x g). Representative Coomassie-stained SDS-PAGE of pellet samples for different combinations of FA proteins with actin, with increasing amounts of PI(4,5)P₂. Graph shows mean densitometry and individual

*Figure 6 continued on next page*

*Figure 6 continued*

values from three independent samples. Related data are shown in *Figure 6—figure supplement 1*. (B) Side-view confocal images of Tn2, Tn2+Vn, and Tn2+Vn[2A] with actin on 5% PI(4,5)P$_2$ supported lipid bilayers after 45 min of actin polymerization, carried out in TIRFm buffer with 15 mM glucose, 20 µg/mL catalase, 100 µg/mL glucose oxidase. Reaction components are as follows: 1 µM actin (10% actin-ATTO488) (green), 0.5 µM Tn2, 0.5 µM Tn2-SNAP647 (blue), 1 µM Vn-SNAP594 or Vn[2A]-SNAP594 (red). Actin bundles appear as bright spots at the membrane surface, indicated by green asterisks for the grayscale actin images. See *Figure 6—figure supplements 2* and *3*, *Figure 6—videos 1* and *2* for additional data from SLB experiments. (C) Comparison of the actin signal at the membrane surface for Tn2, Tn2+Vn, and Tn2+Vn[2A] after 45 min of actin polymerization. Measurements were taken from three independent experiments, at least nine images from each. ****p<0.0005, one-way ANOVA. (D) Top view of SLBs after 45 min of actin polymerization. Projections of 3-slice z-stack at the membrane surface for each condition tested. Scale bar = 10 µm.

The online version of this article includes the following video, source data, and figure supplement(s) for figure 6:

**Figure supplement 1.** Additional cosedimentation experiments corresponding to *Figure 6*.
**Figure supplement 2.** Fluorescence recovery after photobleaching of TMR-PC SLBs, with and without Tn2.
**Figure supplement 2—source data 1.** source data corresponding to *Figure 6—figure supplement 2*.
**Figure supplement 3.** Talin-vinculin interactions regulate actin localization at PIP2-containing SLBs.
**Figure supplement 3—source data 1.** source data corresponding to *Figure 6C* and *Figure 6—figure supplement 3B*.
**Figure 6—video 1.** Side view of actin polymerization in the presence of PIP2-containing SLBs under different conditions.
https://elifesciences.org/articles/56110#fig6video1
**Figure 6—video 2.** Actin organization differs in the presence of talin and vinculin or vinculin[2A].
https://elifesciences.org/articles/56110#fig6video2

exhibit bundles in closer proximity to the membrane, although assessing direct membrane attachment was not possible with the current experimental setup. However, it should be noted that we did not observe specific localization of talin and vinculin at the membrane surface, an important aspect of FA organization in cells, suggesting that additional layers of regulation target talin and vinculin to discrete locations within FAs. Overall, these results reinforce our SLB experiments, and present a direct observation of the membrane-dependent activation of talin and vinculin, leading to actin reorganization.

## Discussion

### Reconstitution of talin-vinculin-actin interactions in vitro

In order to understand how FA interactions are regulated, it is crucial to precisely understand the core interactions of talin and vinculin. Characterizing their mechanisms of regulation has been challenging due to the difficulty of working with the full-length, autoinhibited proteins in vitro. Here, we reconstituted talin-vinculin-actin complexes in the presence of PI(4,5)P$_2$-rich phospholipid bilayers, yielding insight into the regulation of talin-vinculin interactions and revealing a complex web of interdependent regulatory interactions. In the presence of PI(4,5)P$_2$-containing membranes, talin and vinculin have a combined actin binding activity in vitro. Our results suggest that phosphoinositide-mediated activation of talin is sufficient to trigger talin-vinculin interactions, thereby activating vinculin actin binding and recruitment of actin filaments to the membrane (*Figure 8*). Importantly, the reconstitution of stable PI(4,5)P$_2$-talin-vinculin-actin interactions opens the door for further systematic studies of the dynamic interactions between FA proteins, allowing direct observations of key phenomena under simplified, controllable conditions.

We demonstrated for the first time the multivalent interactions between actin, talin, and vinculin. Talin and vinculin are capable of binding to both each other and actin simultaneously; however, it is still unclear if they interact with actin at distinct sites. The vinculin tail domain binds to two perpendicular actin monomers via binding sites along a filament, which is predicted to trigger dimerization of the vinculin tail (*Janssen et al., 2006*). The C-terminal talin actin-binding site, ABS3, has been shown to bind to three actin monomers along a filament, overlapping with the binding site of the vinculin tail (*Gingras et al., 2008*). While it is unlikely that talin and vinculin bind to the same actin monomers, it is plausible that they bind adjacent filaments and cross-link through dimerization. Due to the propensity of both proteins to dimerize, as well as the fact that talin has multiple binding sites for both actin and vinculin, various types of interactions between talin, vinculin, and actin could occur simultaneously. Further structural analysis of talin-vinculin-induced actin bundles is required to clarify the interactions underlying the actin cross-linking activity of talin and vinculin observed here.

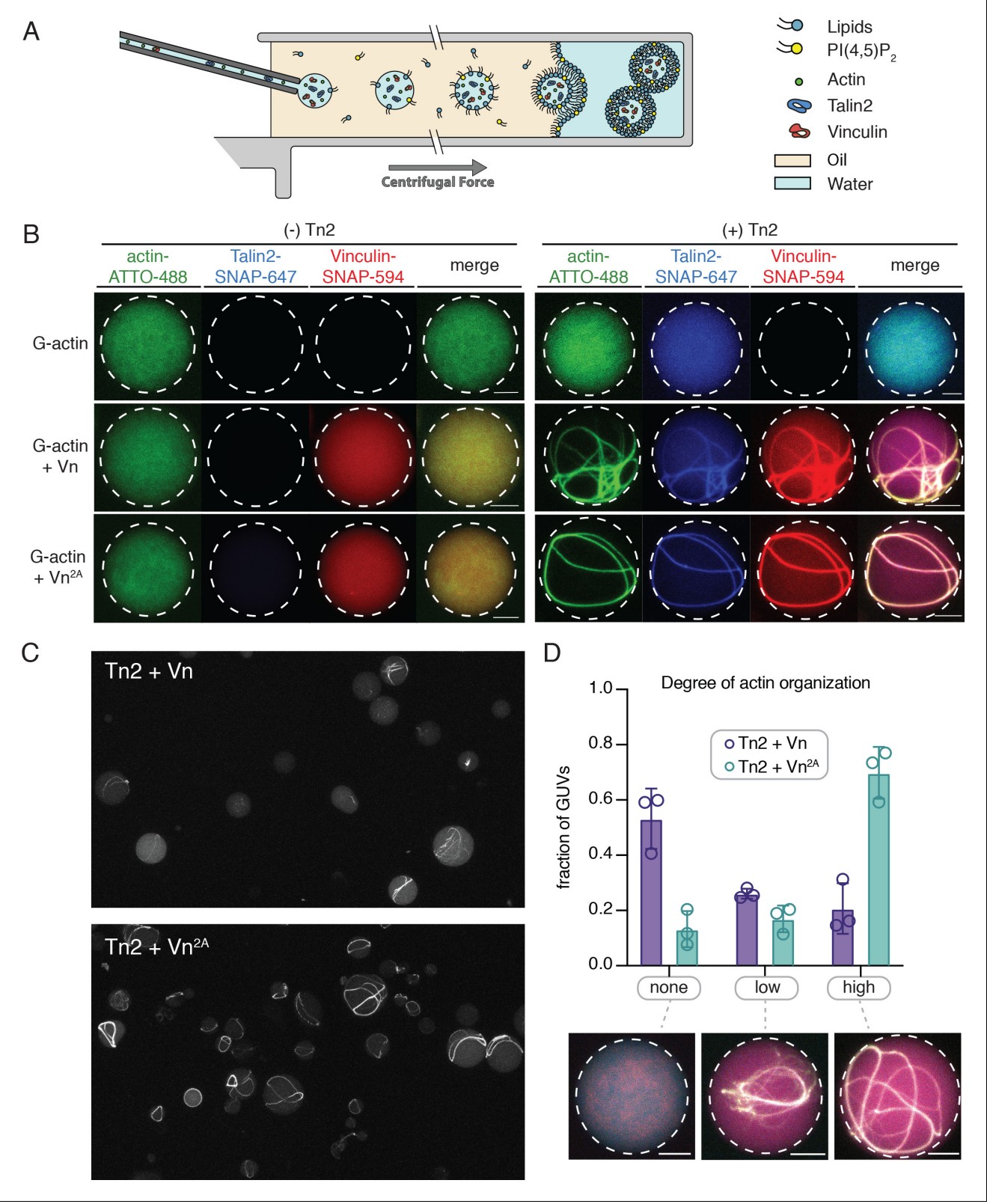

**Figure 7.** Encapsulated talin and vinculin reorganizes F-actin into highly bundled and cross-linked structures. (**A**) Schematic of vesicle encapsulation of FA proteins with cDICE. (**B**) Representative confocal images of individual GUVs (0.975 DOPC, 0.025 PI(4,5)P$_2$ with encapsulated actin, Tn and Vn carried out in TIRFm buffer. Membrane surface indicated by dotted white lines. Scale bars = 5 μM. (**C**) Field of view confocal image of GUVs formed using cDICE, encapsulating Tn2 with Vn or Vn$^{2A}$, and G-actin (10% actin-ATTO488) (2 μM) under polymerizing conditions. (**D**) Comparison of actin

*Figure 7 continued on next page*

*Figure 7 continued*

organization in vesicles with Tn2 and either Vn or Vn[2A]. The number of actin bundles, from three independent experiments for each condition, was used to characterize the degree of actin reorganization within the vesicles, n = 531 (Vn[2A]), 398 (Vn). Scale bars = 2.5 μm. See also *Figure 7—figure supplement 1* and *Figure 7—video 1*.

The online version of this article includes the following video and figure supplement(s) for figure 7:

**Figure supplement 1.** MinDE oscillations indicate successful incorporation of PIP$_2$ in cDICE-generated vesicles.

**Figure 7—video 1.** Rotating 3D views of encapsulated focal adhesion proteins from *Figure 7B*.

https://elifesciences.org/articles/56110#fig7video1

## PI(4,5)P$_2$ drives activation of FA proteins talin and vinculin

Membrane-binding of talin via the FERM domain is sufficient to reveal at least one vinculin-binding site in talin. It is likely that, as predicted by our previous study (*Dedden et al., 2019*), interactions with PI(4,5)P$_2$ release the clamp-like interaction between the FERM domain and R12, subsequently releasing R9, and loosening the autoinhibitory interactions between the rod domains, thus allowing vinculin to bind. Interestingly, removing the FERM domain does not replicate this effect, and is unable to activate wild-type vinculin. This suggests that, in addition to disrupting interactions between the talin head and rod domains, membrane binding may induce a secondary conformational change to reveal a binding site in the talin rod capable of recruiting wild-type, autoinhibited vinculin. Another explanation is that vinculin activation depends on proximity to the membrane surface, requiring the coincidence of PI(4,5)P$_2$-bound talin and direct vinculin-lipid interactions.

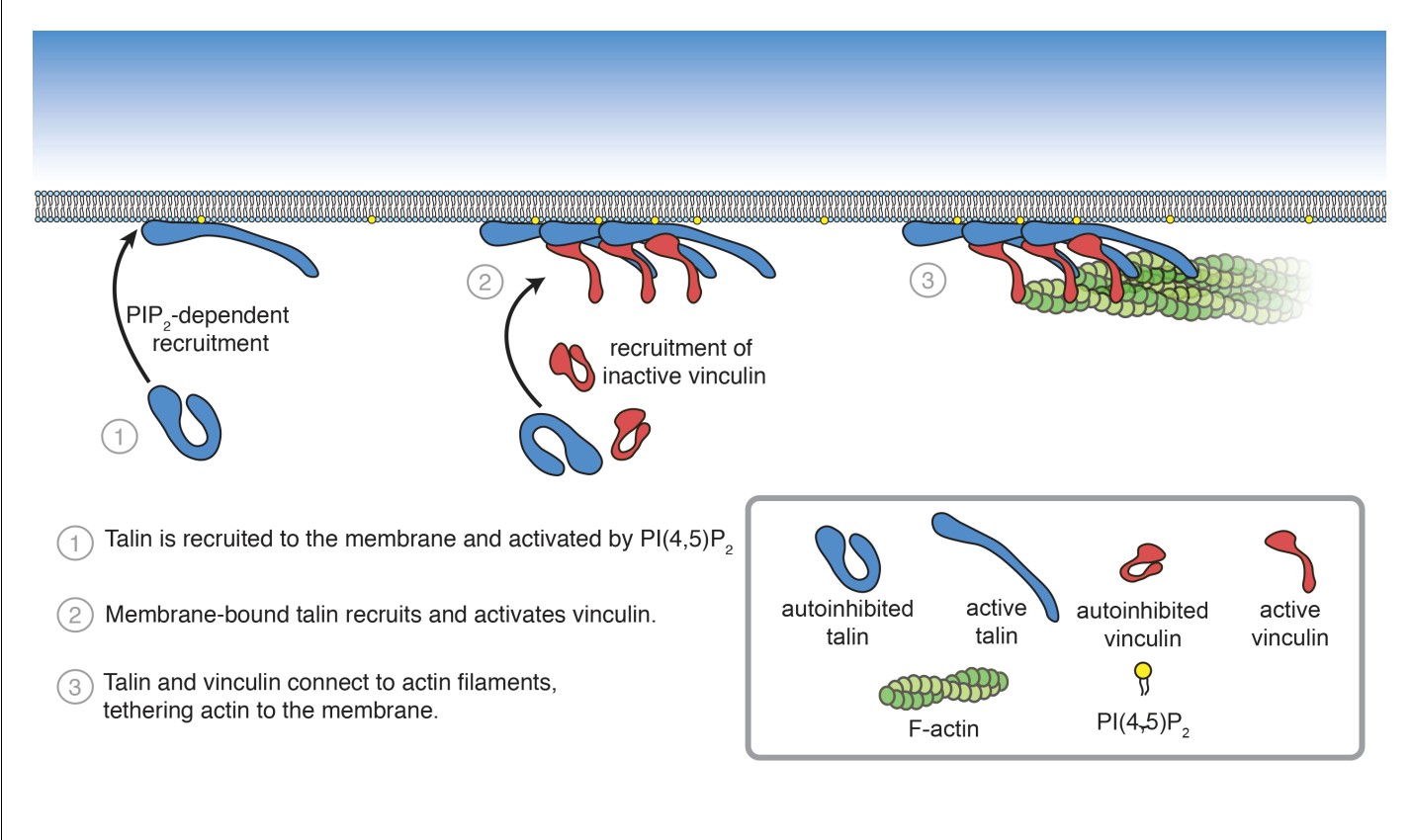

**Figure 8.** Summary of PI(4,5)P$_2$-mediated talin and vinculin activation, and attachment of actin to the membrane surface.

The online version of this article includes the following figure supplement(s) for figure 8:

**Figure supplement 1.** Model of FA assembly.

Interactions between the vinculin tail and acidic phospholipids are predicted to play a role in vinculin regulation (*Johnson and Craig, 1995b*; *Weekes et al., 1996*; *Witt et al., 2004*). Vinculin was found to interact with acidic phospholipids (*Fukami et al., 1994*) through two distinct interfaces in the tail domain (*Bakolitsa et al., 1999*; *Chinthalapudi et al., 2014*; *Thompson et al., 2017*). The basic collar, including the C-terminal tail, is thought to bind specifically to PI(4,5)P$_2$, while the basic ladder, a hydrophobic region located along vinculin tail helix 3, is predicted to insert into the bilayer (*Thompson et al., 2017*). To date, investigations of vinculin-membrane interactions have relied on characterization of the vinculin tail domain in vitro and expression of full-length vinculin mutants in cells. These investigations have led to numerous lipid binding deficient vinculin mutants without identifying a clear, specific lipid-binding site. When expressed in cells, these mutants have a wide range of effects, including changes in focal adhesion size and dynamics, vinculin localization and activation, and actin organization (*Bakolitsa et al., 1999*; *Bakolitsa et al., 2004*; *Chandrasekar et al., 2005*; *Palmer et al., 2009*; *Saunders et al., 2006*; *Steimle et al., 1999*; *Chandrasekar et al., 2005*; *Chinthalapudi et al., 2014*; *Thompson et al., 2017*). Many of these mutations not only reduce membrane binding, but also alter vinculin structure, stability, or actin bundling activities (*Steimle et al., 1999*; *Saunders et al., 2006*; *Palmer et al., 2009*; *Thompson et al., 2017*).

Our results, as well as those presented by Cohen et al., suggest that mutation of lipid-binding residues within the basic ladder (K944, R945) are also key residues in the vinculin head-to-tail autoinhibitory interaction (*Cohen et al., 2005*). These results highlight the importance of considering lipid binding in the context of the full-length protein, and raise questions about the underlying role of vinculin-lipid interactions in FAs. Both the vinculin N773A,E775A and K944Q,R945Q,K1061Q mutants by-pass the requirement for PI(4,5)P$_2$-rich membranes to bind to talin. Interestingly, these mutations also result in a shift in vinculin localization within FAs when expressed in cells, away from the membrane (*Case et al., 2015*; *Thompson et al., 2017*). In our SLB experiments, the loss of autoinhibition results in decreased localization at the membrane surface and an increase in actin bundling (*Figure 6*). These results do not determine whether direct membrane interactions are required for localization or activation of wild-type vinculin, as loss of autoinhibition may simply circumvent this requirement, but indicate a close link between vinculin lipid binding and autoregulation. Further investigation is needed to determine whether a direct vinculin-lipid interaction can in fact be detected in vitro, and what role this interaction plays in regulating vinculin activity within FAs.

## Intramolecular breathing may contribute to vinculin-talin-actin interactions

Our experiments revealed interdependent regulatory interactions, which can be summarized by two major observations: (i) partial disruption of vinculin head-to-tail interactions reveals the actin binding and reorganizing activities of talin and vinculin, and (ii) phosphoinositide-mediated activation of talin leads to talin-vinculin interactions and recruitment of actin filaments. These findings indicate that release of autoinhibition of either protein is sufficient to engage the other, consistent with recent results of a mitochondrial targeting assay in cells (*Atherton et al., 2020*). It also highlights that talin-vinculin interactions are not a simple case of 'open-closed' regulation. In the case of partially deregulated vinculin (i), the coincidence of talin, vinculin, and actin results in stable interactions, while none of the binary interactions are detected in isolation. This paradox can be explained by intramolecular breathing, whereby structural elements undergo small-scale stochastic fluctuations (*Kossiakoff, 1986*; *Makowski et al., 2008*) that allow contact points to open up, increasing the chance that secondary effects will take place. This can apply to both vinculin and talin, in which case fluctuations of autoinhibitory interactions would briefly reveal binding sites (*Stutchbury, 2016*). In the presence of a single ligand, the probability is low that an interaction will be strong enough to disrupt both autoinhibitory interactions to form a stable complex, but in the presence of an additional ligand, such as actin, the probability of simultaneously interrupting both autoinhibitory intramolecular interactions significantly increases, resulting in an activated complex. Activation of membrane-bound talin (ii) is also not as straightforward as disrupting a single interaction interface. If this were the case, simply removing the FERM domain (Tn$^{\Delta Head}$) should activate wild-type vinculin. Instead there is well tuned and synergistic effect from both proteins. Interestingly, a recent study reported that a fragment similar to Tn$^{\Delta Head}$, was able to bind to wild-type vinculin in cells, suggesting that under physiological conditions the vinculin-binding sites in the talin rod are more available, or that another cellular factor, aside from PI(4,5)P$_2$, can mediate initial talin-vinculin interactions in the absence of

the FERM domain (*Atherton et al., 2020*). Overall, these observations shed light on the multifaceted mechanisms regulating vinculin-talin-actin interactions, which likely play an important role in FA assembly and dynamics.

## A role for PI(4,5)P$_2$ in FA assembly

The role of force transduction and actomyosin-mediated tension in FA assembly and maturation has been the focus of intense interest in recent years. Talin and vinculin form the core mechanosensitive machinery in FAs, directly connecting integrin receptors to the actomyosin network. Both proteins are under tension when engaged at FAs (*Grashoff et al., 2010*; *Austen et al., 2015*; *Kumar et al., 2016*; *LaCroix et al., 2018*), and force on talin reveals additional vinculin-binding sites in the talin rod (*del Rio et al., 2009*, *Yao et al., 2015*). These findings led to the hypothesis that force is required for talin activation, vinculin-talin interactions, and FA maturation. However, recent studies by Dedden et al. and Atherton et al. showed that force-independent interactions between talin and vinculin can occur if autoinhibition of either protein is relieved. Here, we demonstrate that binding to PI(4,5)P$_2$-rich membranes can initiate force-independent interactions between talin and vinculin, and subsequently link actin to the membrane.

How would this mechanism of activation specifically target talin-vinculin-actin interactions to sites of adhesion initiation and assembly? It is likely that membrane binding is downstream of initial recruitment events. Talin localizes specifically to FAs, and does not diffuse laterally on the membrane (*Rossier et al., 2012*), suggesting additional lipid-independent recruitment mechanisms, possibly mediated by interactions with the small GTPase Rap1 and its effector RIAM. Rap1-RIAM-talin interactions coordinate integrin activation, may enhance talin interactions with the membrane, and likely precede talin activation and vinculin interactions (*Goult et al., 2013b*; *Yang et al., 2014*; *Gingras et al., 2019*; *Bromberger et al., 2019*). Significantly, talin also binds and activates PIPK1γ, the kinase responsible for regulating PI(4,5)P$_2$-levels at FAs (*Legate et al., 2011*), via the F3 subdomain (*Ling et al., 2002*; *Di Paolo et al., 2002*). As a result, once talin is localized at the membrane, talin-PIPK1γ interactions could induce local increases in PI(4,5)P$_2$ levels, initiating a positive feedback loop leading to additional talin recruitment at sites of FA initiation. Once engaged at the membrane, the interactions between the talin head and rod domains are released, allowing integrin and other proteins, including vinculin, to bind. Vinculin is likely recruited to FAs by paxillin, another FA adaptor protein, and then activated by talin (*Case et al., 2015*). Once active, talin and vinculin attach to the actin cytoskeleton, bringing together the core mechanosensitive FA machinery. Actomyosin driven forces then place the entire assembly under tension, leading to further protein recruitment, strengthening the connection between integrin and the actin cytoskeleton, and ultimately leading to FA maturation and stability (*Stutchbury, 2016*; *Atherton et al., 2020*; *Figure 8—figure supplement 1*).

PIP$_2$ levels are also implicated in regulating focal adhesion disassembly (*Di Paolo et al., 2002*; *Ling et al., 2002*, *D'Souza et al., 2020*). The balance of interactions between talin, PIPK1γ, integrin, vinculin, and PI(4,5)P$_2$ likely play a central role in regulating both focal adhesion formation and turnover. Moving forward, it will be important to look closely at the dynamic interactions between these proteins and phospholipids, as well as the underlying signaling events that regulate them, in order to understand spatial and temporal regulation of FAs.

# Materials and methods

**Key resources table**

| Reagent type (species) or resource | Designation | Source or reference | Identifiers | Additional information |
|---|---|---|---|---|
| Gene (*Homo sapiens*) | TLN2 | Human Genome Nomenclature Database | HGNC:15447 | |
| Gene (*Homo sapiens*) | VCL | Human Genome Nomenclature Database | HGNC:12665 | |

*Continued on next page*

Continued

| Reagent type (species) or resource | Designation | Source or reference | Identifiers | Additional information |
|---|---|---|---|---|
| Strain, strain background (*Escherichia coli*) | BL21-Gold (DE3) | Agilant | Cat# 230132 | |
| Strain, strain background (*Escherichia coli*) | XL1-Blue | Agilant | Cat# 200249 | |
| Recombinant DNA reagent | pCB-a-bax33-hsTln2-3C-SNAP-6His | This paper, Subcloned from gift from Carsten Grashoff | | includes L435G mutation to reduce calpain cleavage |
| Recombinant DNA reagent | pCB-a-bax33-hsTln2$^{ABS2}$-3C-SNAP-6His | This paper | | L435G, K925E, Q926E, Q933E, R1502E, R1512E, K1524E |
| Recombinant DNA reagent | pCB-a-bax33-hsTln2$^{ABS3}$-3C-SNAP-6His | This paper | | L435G, K2244D, K2245D, K2246D |
| Recombinant DNA reagent | pCB-a-bax33-hsTln2$^{ABS2/3}$-3C-SNAP-6His | This paper | | L435G, K925E, Q926E, Q933E, R1502E, R1512E, K1524E, K2244D, K2245D, K2246D |
| Recombinant DNA reagent | pCB-a-bax33-hsTln2$^{N:R3}$-3C-SNAP-6His | This paper | | Tln2(1-910) with L435G mutation |
| Recombinant DNA reagent | pCB-a-bax33-hsVcl-3C-SNAP-6His | This paper, Subcloned from gift from Carsten Grashoff | | |
| Recombinant DNA reagent | pCB-a-bax33-hsVcl$^{2A}$-3C-SNAP-6His | This paper | | N773A,E775A |
| Recombinant DNA reagent | pCB-a-bax33-hsVcl$^{D1}$-3C-SNAP-6His | This paper | | Vcl(1-258) |
| Rcombinant DNA reagent | pCB-a-bax33-hsVcl$^{Head}$-3C-SNAP-6His | This paper | | Vcl(1-823) |
| Recombinant DNA reagent | pCB-a-bax33-hsVcl$^{Tali}$-3C-SNAP-6His | This paper | | Vcl(878–1066) |
| Recombinant DNA reagent | pCB-a-bax33-hsVcl$^{Tail3Q}$-3C-SNAP-6His | This paper | | Vcl(878–1066) K944Q, R944Q, K1061Q |
| Recombinant DNA reagent | pCB-a-bax33-hsVcl$^{3Q}$-3C-SNAP-6His | This paper | | K944Q, R945Q, K1061Q |
| Peptide, recombinant protein | 3C-6his protease | This lab | | Purified from *E. coli* BL21-Gold (DE3) |
| Peptide, recombinant protein | Actin Rabbit skeletal muscle alpha actin | Hypermol | Cat# 8101–01 | |
| Peptide, recombinant protein | Actin-ATTO488 Rabbit skeletal muscle alpha actin | Hypermol | Cat# 8153–02 | |
| Peptide, recombinant protein | Actin-ATTO488-TIRF Rabbit skeletal muscle alpha actin | Hypermol | Cat# 8153–03 | |
| Peptide, recombinant protein | Actin-biotin Rabbit skeletal muscle alpha actin | Hypermol | Cat# 8109–01 | |
| Peptide, recombinant protein | VBS (vinculin activating peptide) | Hypermol | Cat# 8317–01 | Derived from C-terminal talin sequence |

*Continued on next page*

*Continued*

| Reagent type (species) or resource | Designation | Source or reference | Identifiers | Additional information |
|---|---|---|---|---|
| Peptide, recombinant protein | NeutrAvidin | Thermo Fischer Scientific | 31000 | |
| Chemical compound, drug | mPEG-silane, MW 2000 | Laysan Bio Inc | Cat# MPEG-SIL-2000–1 g | |
| Chemical compound, drug | mPEG-silane-biotin. MW 3400 | Laysan Bio Inc | Cat# Biotin-PEG-SIL-3400–1 g | |
| Chemical compound, drug | Methyl Cellulose | Acros Organics | Cat# 10287200 | viscosity 4000 c |
| Peptide, recombinant protein | Glucose Oxidase Type II | Sigma Aldrich Chemie GmbH | G6125 | From aspergillus |
| Peptide, recombinant protein | Catalase | Sigma Aldrich Chemie GmbH | E3289 | From bovine liver |
| Software, algorithm | FIJI | | https://imagej.net/Fiji/Downloads | |
| Software, algorithm | Prism | Graphpad | https://www.graphpad.com/scientific-software/prism/ | |
| Other | DOPC (18:1 (Δ9-Cis) PC) | Avanti Polar Lipids | Cat# 850375 | 1,2-dioleoyl-sn-glycero-3-phosphocholine |
| Other | DOPE (18:1 (Δ9-Cis) PE) | Avanti Polar Lipids | Cat# 850725 | 1,2-dioleoyl-sn-glycero-3-phosphoethanolamine |
| Other | DOPS (18:1 PS) | Avanti Polar Lipids | Cat# 840035 | 1,2-dioleoyl-sn-glycero-3-phospho-L-serine (sodium salt) |
| Other | TopFluor TMR PC | Avanti Polar Lipids | Cat# 810180 | 1-oleoyl-2-(6-((4,4-difluoro-1,3-dimethyl-5-(4-methoxyphenyl)-4-bora-3a,4a-diaza-s-indacene-2-propionyl)amino)hexanoyl)-sn-glycero-3-phosphocholine |
| Other | Brain PI(4,5)P$_2$ | Avanti Polar Lipids | Cat# 840046 | L-α-phosphatidylinositol-4,5-bisphosphate (Brain, Porcine) (ammonium salt) |
| Other | 18:0-20:4 PI(4,5)P$_2$ synthetic | Avanti Polar Lipids | Cat# 850165 | 1-stearoyl-2-arachidonoyl-sn-glycero-3-phospho-(1'-myo-inositol-4',5'-bisphosphate) |
| Other | sticky-Slide VI 0.4 | Ibidi | Cat# 80608 | |

## Plasmids

All plasmids used for expressing the proteins in this study were generated specifically for this study. To generate plasmids for expressing hsTalin2 (includes L435G mutation to reduce calpain cleavage), hsVinculin, and hsVinculinN773A,E775A as his-SNAP-tagged fusions in *E. coli*, ORFs were PCR amplified from pCB-hsTalin2-eGFP-his8 and pEC-A-3C-His-GST-hsVinculin, pEC-A-3C-GST-hsVinculinN773A,E775A (*Dedden et al., 2019*) and assembled using Gibson Assembly (*Gibson et al., 2009*). Vinculin truncations were amplified from pCB-hsVinculin-SNAP-8his by PCR and assembled using Gibson assembly. To generate talin2 ABS mutant plasmids, gene fragments containing mutations were ordered from Twist Bioscience (San Francisco, California), and assembled with fragments amplified from the wild-type pCB-hsTalin2-SNAP-his8 plasmid. Site-directed mutagenesis was

performed on pCB-hsVinculin-SNAP-his8 to generate the lipid binding mutant pCB-hsVinculin (K944Q,R945Q,K1061Q)-SNAP-his8.

## Protein expression and purification

Constructs were expressed in *E. coli* BL21 (DE3) gold using ZY auto-induction medium. Talin proteins were all purified using the same protocol, based on that described in a previous report (*Dedden et al., 2019*). Cells were lysed by sonication in 50 mM Tris-HCl pH 7.8, 500 mM NaCl, 5 mM imidazole, 3 mM β-mercaptoethanol, 1 mM EDTA, and Roche cOmplete protease inhibitor tablets, followed by purification using nickel-affinity chromatography (cOmplete His-Tag purification column, Roche), and cation (HiTrap SP FF, GE Healthcare) exchange. Next, the his-tag was either removed using overnight incubation with 3C protease, or labeled using overnight incubation with SNAP-AlexaFluor647 (New England Biolabs, Ipswich, Massachusetts). Finally, protein was further purified by size-exclusion chromatography using either a Superdex 200 16/600 column (GE Healthcare) or Superose 6 10/300 column (GE Healthcare) in 50 mM HEPES pH 7.8, 150 mM KCl, 3 mM β-mercaptoethanol, 1 mM EDTA, and 10% glycerol, followed by flash freezing for storage at −80°C.

Vinculin proteins were lysed by sonication in 50 mM Tris-HCl pH 7.8, 500 mM NaCl, 5 mM imidazole, 3 mM β-mercaptoethanol, 1 mM EDTA, and Roche cOmplete protease inhibitor tablets (Roche, Basil, Switzerland). Following lysis, TritonX-100 was added for a final amount of 1% by volume. Full-length vinculin cell lysates were incubated on Roche cOmplete His-Tag resin for 2 hr at 4°C, then washed with 50 mM Tris-HCl pH 7.8, 500 mM NaCl, 10 mM imidazole, 3 mM β-mercaptoethanol, 1 mM EDTA. After washing, proteins were incubated overnight with either 3C protease to remove the SNAP-his tag, or labeled with SNAP-AlexaFluor488 or SNAP-Surface594 (New England Biolabs). Following removal or elution from beads, vinculin proteins were then further purified by size-exclusion chromatography using Superdex 200 16/600 column (GE Healthcare) or Superose 6 10/300 column (GE Healthcare) in 50 mM HEPES pH 7.8, 150 mM KCl, 3 mM β-mercaptoethanol, 1 mM EDTA. Vinculin fragments were purified using nickel-affinity chromatography, immediately eluted from the column with 1M imidazole, cleaved overnight with 3C, and separated from the cleaved SNAP-his tag by reverse nickel-affinity chromatography. This was followed by size-exclusion chromatography using a Superdex 75 10/300 in 50 mM HEPES pH 7.8, 150 mM KCl, 3 mM β-mercaptoethanol, 1 mM EDTA. Proteins were flash frozen and stored at −80°C.

The VBS (vinculin-binding site) peptide (Hypermol, Bielefeld, German) used in this study is a 2.9 kDa peptide derived from a C-terminal vinculin-binding site in talin.

## Analytical size-exclusion chromatography assays

Proteins used were first buffer exchanged or diluted to match the conditions tested (i.e. 75 mM or 500 mM KCl). Proteins were incubated together on ice for 15 min, and prespun in a tabletop microcentrifuge at 15,000 x g. 75% of the sample volume was removed, avoiding any potential pellet, and applied to a Superose 6 Increase 3.2/300 column with 20 mM HEPES pH 7.8, 1 mM EDTA, 3 mM β-mercaptoethanol, and either 75 mM or 500 mM KCl.

## Actin co-sedimentation assays

In all cases, actin (rabbit skeletal alpha actin, Hypermol, Bielefeld, Germany) was resuspended in deionized water and pre-spun at 15,000 x g for 10 min at 4°C in a tabletop microcentrifuge. The top 75% of the actin solution was then transferred to a new Eppendorf tube and kept on ice for the duration of the experiment. Proteins were prepared and buffer adjusted to attain a final reaction concentration of 75 mM KCl. Premixed proteins were prespun either at 100,000 x g (actin pelleting assays) or 10,000 x g (actin bundling assays) in a Beckman Optima Max XP tabletop ultracentrifuge and TLA 100 rotor using Beckman Coulter polycarbonate centrifuge tubes to remove potential aggregates (Beckman Coulter, Brea, California). 80% of the supernatant was then transferred to a tube and stored on ice for the duration of the experiment.

Actin in G-buffer was polymerized with the addition of 50 mM KCl, 2 mM MgCl$_2$, and 0.2 mM ATP for 15 min at room temperature. The indicated proteins were then added to polymerized actin and incubated for an additional 30 min at room temperature. This was followed by centrifugation at either 100,000 x g (F-actin pelleting) or 10,000 x g (F-actin bundle pelleting). After the spin, 75% of the supernatant was removed and mixed with SDS sample buffer. The remainder of the supernatant

was discarded, and the pellet resuspended in sample buffer. Equal volumes of pellet and supernatant were analyzed by gradient SDS-PAGE, and quantified using Fiji.

### Dynamic light-scattering (DLS)

DLS was carried out as described previously (*Dedden et al., 2019*). Briefly, talin2 was prepared in 20 mM HEPES pH 7.5, 75 mM KCl, 1 mM EDTA, 3 mM 3 mM β-mercaptoethanol and pre-spun at 18,000 x g. Samples were diluted to a final Tn2 concentration of 0.3 mg/mL and KCl concentrations ranging from 75 to 500 mM. Using a Dynapro Platereader-III Dynamic Light Scattering instrument (Wyatt Technology Corporation) at 20°C in a 96-well plate, four measurements per well were taken for three independent measurements of each sample. The data was analyzed in Dynamic 7.8 (Wyatt Technology Corporation) and plotted with PRISM (GraphPad). The fitting of the curve was performed using a nonlinear regression fit with a sigmoidal curve.

### Total internal reflection fluorescence (TIRF) microscopy

For all TIRF experiments, 24 × 60 mm coverslips were cleaned as described previously (*Jansen et al., 2015*). Briefly, coverslips sonicated for 1 hr in detergent, 20 min in 1 M KOH, 20 min in 1 M HCl, and 1 hr in EtOH. In between each successive sonication, coverslips were washed 7x with ddH20. Coverslips were then stored in EtOH for up to 3 days. Clean coverslips were washed extensively with ddH20 and dried with $N_2$, then coated using a solution of 80% ethanol pH 2, 2 mg/mL methoxy-poly (ethylene glycol)-silane and 2 ug/mL biotin-poly (ethylene glycol)-silane (Laysan Bio Inc, Arab, Alabama) and incubated at 75°C for up to 2 days. Immediately before using, coverslips were washed extensively with ddH20, dried under a stream of $N_2$, and attached to adherent flow chambers (Ibidi, Martensried, Germany).

Slide chambers were pre-treated as follows: 80 µl each of 1 mg/mL BSA (3 min incubation), 0.1 mg/mL Neutravidin in 10 mM Tris pH 8 (1 min incubation), and twice rinsed with TIRFm buffer (10 mM imidazole, 50 mM KCl, 1 mM MgCl2, 1 mM EGTA, 0.2 mM ATP, pH 7.5). Reaction mixes containing FA proteins and actin (5% ATTO488-actin for TIRFm, 0.5% biotinylated-actin) in TIRFm buffer supplemented with 15 mM glucose, 20 µg/mL catalase, 100 µg/mL glucose oxidase, 1 mM DTT and 0.25% methyl-cellulose (4000 cp).

Single-color TIRFm was carried out on a GE DeltaVision with an Olympus UAPON TIRF 100x/1.49 objective, using softWoRx imaging software with an Ultimate Focus module. Labeled actin (Hypermol) was mixed with untagged FA proteins and immediately transferred to flow chamber. Timelapses were recorded for first 15 min of polymerization, with images captured every 15 s. Multi-color TIRF was carried out using a Zeiss (Oberkochen, Germany) Elyra PS.1 with an alpha Plan-Apochromat 100x/1.46 oil objective, CMOS camera, and using Zeiss Zen Black with a Definite Focus module. Labeled actin was mixed with SNAP-tagged FA proteins and immediately transferred to a flow chamber.

Analysis was carried out in Fiji, using the MorpholibJ plugin (*Legland et al., 2016*) to determine fraction of field of view containing signal from actin filaments. The number of filaments per bundle was estimated by the average fluorescence of single actin filaments from actin control images. The peak intensity across individual filaments/bundles for different conditions was measured using Fiji and divided by the average signal from a single filament to give an estimate of filament number.

### Lipid co-sedimentation assays

Lipid co-sedimentation assay were conducted as previously described (*Becalska et al., 2013*; *Kelley et al., 2015*). Briefly, liposomes were swelled from dried lipid in 20 mM HEPES, pH 7.5 and 100 mM NaCl. FA proteins were mixed with 1 mg/mL liposomes and incubated at room temperature for 30 min, then spun at 18000 x g in a tabletop microcentrifuge at 4°C. Equal volumes of pellet and supernatant were analyzed by gradient SDS-PAGE, and quantified using Fiji. For quantification, the percent of protein in the pellet of protein-alone control samples was subtracted from all experimental samples.

### GUV assays

GUVs were generated using the gentle hydration method. Briefly, 1 mg of lipids (composition indicated in figure legends) were mixed in 20:9:1 chloroform:methanol:ddH$_2$O mix, dried under a

continuous stream of $N_2$, and further dried under vacuum for 2 hr at room temperature. Lipids were then rehydrated in 2 mL of 300 mM sucrose pH 7.5 overnight at 75°C. The GUV cloud was transferred to a new Eppendorf tube and resuspended in GUV imaging buffer (5 mM HEPES pH 7.5, 100 mM KCl), diluted by a factor of 200.

Ibidi slide chambers were precoated with 1 mg/mL BSA. GUVs and FA proteins were mixed, immediately transferred to the slide chamber, and incubated for 30 min at room temperature. GUVs were then imaged with a Leica (Wetzlar, Germany) SP8 laser scanning confocal with a HC PL APO 63x/1.40 oil objective, using LASX imaging software. Only unilamellar GUVs were quantified, and quantification was carried out using Fiji.

Continuous droplet interface crossing encapsulation (cDICE) assays was carried out as described previously (*Litschel et al., 2018*). Briefly, 5% ATTO488-actin was prepared similar to that used for TIRFm. FA proteins were premixed, using 25% Tn-SNAP647, and 75% unlabeled Tn, and 100% Vn-SNAP594. Immediately before loaded into cDICE setup, proteins were diluted into F-buffer. The vesicles were pipetted into a microtiter plate for imaging (Greiner Bio-One, 96-well glass bottom SensoPlateTM), passivated beforehand with 50 µl of 5 mg/mL casein for 20 min. Imaging was performed with a Zeiss LSM 780/CC3 confocal microscope equipped with a C-Apochromat, 63x/1.4 W objective. PMT detectors (integration mode) were used to detect fluorescence emission (excitation at 488 nm for ATTO488, 594 nm for SNAP594 and 633 nm for SNAP647) and record confocal images. All experiments were conducted at room temperature.

## Supported lipid bilayer assays

The SLB protocol used was adapted from that described previously (*Braunger et al., 2013*). Briefly, lipids were mixed in a glass vial, and dried under a continuous stream of $N_2$, then dried overnight in a vacuum chamber at room temperature. The lipid film is then gently rehydrated in citric acid buffer pH 4.8, and incubated at room temperature for 20 min before vortexing briefly. To produce small unilamellar vesicles (SUVs), the solution is then sonicated for 30 min (30 s on/30 s off).

Coverslips were cleaned using Piranha Solution for at least 15 min. Immediately before using, coverslips were rinsed thoroughly in deionized water and dried with $N_2$, then attached to Ibidi adherent flow chambers. To form SLBs, 60 µL of 0.2 mg/mL SUVs were added to individual flow chambers and incubated for 3 min, followed by 2 × 80 µl TIRFm buffer (1x) to remove excess vesicles.

SLBs were preincubated with 500 nM Tn2 for 15 min, then tested for bilayer integrity using fluorescence recovery after photobleaching (FRAP). To fluid bilayers, additional FA proteins and actin (5% ATTO488-actin) were added in TIRFm buffer supplemented with GODCAT. Timelapses were acquired for 40 min to allow actin polymerization, followed by z-stacks and side views of the samples with a Leica SP8 laser scanning confocal with a HC PL APO 63x/1.40 oil objective, using LASX imaging software. For quantification in *Figure 6C* and *Figure 6—figure supplement 3*, the fluorescence intensity along the z-axis was averaged along the x-axis for each channel for nine separate images from an individual experiment.

## Abbreviations

FA, Focal Adhesion; FERM, 4.1-ezrin-radixinmoesin; PI(4,5)P2, phosphoinositol-4,5-bisphosphate; Tn, talin; Vn, vinculin; total internal reflection fluorescence microscopy, TIRFm; DLS, dynamic light-scattering; ABS, actin binding site; VBS, vinculin binding site; GUV, giant unilamellar vesicles; SLB, supported lipid bilayer; cDICE, continuous droplet interface crossing encapsulation; DOPC, 1,2-dioleoyl-sn-glycero-3-phosphocholine; DOPE, 1,2-dioleoyl-sn-glycero-3-phosphoethanolamine; DOPS, 1,2-dioleoyl-sn-glycero-3-phospho-L-serine.

## Acknowledgements

We are grateful to Carsten Grashoff for sharing the original hsTalin2 construct, Gijsje Koenderink and Agata Szuba for sharing their PIP$_2$-SLB protocol, Stephan Uebal for DLS measurements, Elena Conti and the Max Planck Institute of Biochemistry light microscopy and biochemistry core facilities, especially, for support and infrastructure; Christian Biertümpfel, Nirakar Basnet, and Mizuno lab members for helpful discussions. We thank Avital Rodal for comments on the manuscript.

# Additional information

## Funding

| Funder | Grant reference number | Author |
|---|---|---|
| European Molecular Biology Organization | EMBO Long-term Fellowship Award | Charlotte F Kelley |
| Alexander von Humboldt-Stiftung | Research Fellowship for Postdoctoral Researchers | Charlotte F Kelley |
| Horizon 2020 Framework Programme | Marie Sklodowska-Curie Action | Charlotte F Kelley |
| Boehringer Ingelheim Stiftung | Plus 3 | Naoko Mizuno |
| H2020 European Research Council | FocAd | Naoko Mizuno |
| European Molecular Biology Organization | Young Investigator Award | Naoko Mizuno |
| Max-Planck-Gesellschaft | | Naoko Mizuno |

The funders had no role in study design, data collection and interpretation, or the decision to submit the work for publication.

## Author contributions

Charlotte F Kelley, Conceptualization, Formal analysis, Funding acquisition, Investigation, Methodology, Writing - original draft, Project administration, Writing - review and editing; Thomas Litschel, Formal analysis, Investigation, Methodology; Stephanie Schumacher, Dirk Dedden, Methodology, Writing - review and editing; Petra Schwille, Resources, Supervision; Naoko Mizuno, Conceptualization, Resources, Supervision, Funding acquisition, Project administration, Writing - review and editing

## Author ORCIDs

Charlotte F Kelley https://orcid.org/0000-0002-7684-9049
Thomas Litschel https://orcid.org/0000-0001-7123-8364
Dirk Dedden http://orcid.org/0000-0003-3630-1270
Petra Schwille http://orcid.org/0000-0002-6106-4847
Naoko Mizuno https://orcid.org/0000-0002-1594-2821

## Decision letter and Author response

Decision letter https://doi.org/10.7554/eLife.56110.sa1
Author response https://doi.org/10.7554/eLife.56110.sa2

# Additional files

## Supplementary files

• Transparent reporting form

## Data availability

All data generated or analysed during this study are included in the manuscript and supporting files. Source data files would be provided for Figures 1, 2, 3, 4 and 6.

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
