## [Decision Letter]

**Acceptance summary:**

Talin, vinculin, and actin are key components of cell-matrix adhesions, but the precise mechanisms by which activities of these proteins are regulated during formation of focal adhesions are incompletely understood. By applying a combination of biochemical and in vitro reconstitution approaches Kelley et al. revealed that interactions between talin, viculin and actin do not represent simple 'open-to-closed' regulation, but that their interplay is mechanistically more complex. Moreover, they elucidated the role of phosphoinositides in the activation of talin and vinculin. Therefore, this study provides important new information on the interplay between talin, vinculin, and phosphoinositides.

**Decision letter after peer review:**

Thank you for submitting your article "Phosphoinositides regulate force-independent interactions between talin, vinculin, and actin" for consideration by *eLife*. Your article has been reviewed by three peer reviewers, including Pekka Lappalainen as the Reviewing Editor and Reviewer #1, and the evaluation has been overseen by Vivek Malhotra as the Senior Editor.

The reviewers have discussed the reviews with one another and the Reviewing Editor has drafted this decision to help you prepare a revised submission. In recognition of the fact that revisions may take longer than the two months we typically allow, until the research enterprise restarts in full, we will give authors as much time as they need to submit revised manuscripts.

The talin-vinculin interaction and its regulation by force or through other activation mechanisms has received considerable interest recently. Here, Kelley et al. applied a combination of biochemical assays and elegant in vitro reconstitution approaches to elucidate the mechanisms by which activities of these proteins are regulated during the assembly of focal adhesions. Importantly, they revealed that interactions between talin, viculin and actin do not represent simple 'open-to-closed' regulation, but instead their interplay is mechanistically more complex. Moreover, they elucidated the role of PI(4,5)P_2_ in activation of talin and vinculin.

All three reviewers stated that his work is of high quality and brings important conclusions concerning the mechanism by which an essential machinery of focal adhesions assembles at the membrane and re-organizes the actin cytoskeleton. However, the reviewers identified few points that should be addressed to further strengthen the study.

Essential revisions:

1) Key controls are missing from some experiments. In Figure 5B and Figure 5—figure supplement 1, the authors should also show what is the proportion of Vn-tail and Vn^tail3Q^ sedimenting in the absence of liposomes. Moreover, the fact that a large fraction of talin (Tn2) is present in the pellet fraction also in the absence of F-actin is a concern (Figure 1—figure supplement 2). The authors should explain the reason for this observation (aggregation of talin?) and discuss the potential consequences of this issue for the interpretation of other experiments where this protein was used. In general, the authors should perform co-sedimentation experiments in all relevant cases also in the absence of F-actin, and compare the results to the ones in the presence of F-actin.

2) Although encapsulation of talin, vinculin and actin in GUVs is an impressive accomplishment (Figure 7), the conclusions of such a beautiful reconstitution are somewhat limited, mainly because the experiment is not sufficiently documented. The authors conclude that "these results reinforce our observations and present a direct observation of the membrane-dependent activation of talin and vinculin, leading to actin reorganization". Actually, to reach this conclusion the respective localizations of talin, vinculin and actin bundles inside the GUVs should be examined. Are talin and vinculin bound to the membrane, like in focal adhesions, or do they accumulate along the actin filament bundles? Are actin bundles floating inside GUVs without anchoring points, or bound to the inner leaflet of the membrane? These issues should be addressed through more thorough analysis of the GUV experiment.

3) The possible role of vinculin-membrane interactions remains obscure. The authors demonstrate that removing the talin FREM domain is not sufficient to activate wild-type vinculin. Since the authors could achieve activation of wild-type vinculin only on PIP_2_-rich membranes (in the presence of talin), it would be interesting to examine, or at minimum to better discuss, the possible role of vinculin – PIP_2_ interactions, or some currently uncharacterized effects of PIP_2_ on talin conformation, in this process.

---

## [Author Response]

Essential revisions:1) Key controls are missing from some experiments. In Figure 5B and Figure 5—figure supplement 1, the authors should also show what is the proportion of Vn^tail^ and Vn^tail3Q^ sedimenting in the absence of liposomes. Moreover, the fact that a large fraction of talin (Tn2) is present in the pellet fraction also in the absence of F-actin is a concern (Figure 1—figure supplement 2). The authors should explain the reason for this observation (aggregation of talin?) and discuss the potential consequences of this issue for the interpretation of other experiments where this protein was used. In general, the authors should perform co-sedimentation experiments in all relevant cases also in the absence of F-actin, and compare the results to the ones in the presence of F-actin.

We thank the reviewer for the suggestion, we have added control gels for the cosedimentation experiments. In all cases, the co-sedimentation experiments were also performed in the absence of actin and in the absence of liposomes. They have been added to the supplementary figures for Figures 2, 3, 5, and 6 to include (-) actin or liposomes control samples where appropriate. We agree that the presence of talin in the pellet sample after high-speed centrifugation (90,000 x g) should be addressed. We have added an additional discussion of this to Figure 1—figure supplement 1:

“We note that talin2 sedimented in the absence of actin. […] Based on these results, as well as the SEC results of Figure 1—figure supplements 1 and 2, Tn2 behaves similar to Tn1 and, thus, is well-folded and functional.”

To clarify for the reviewers, SEC analysis of talin2 shows that its elution profile shifts depending on salt concentrations (75 mM KCl, Figure 1—figure supplement 1, and 500 mM KCl, Figure 1—figure supplement 2), and DLS analysis shows the change of hydrodynamic radius depending on the salt concentration (Figure 1—figure supplement 3). This indicates that talin2 undergoes a conformational change dependent on ionic strength, which has been characterized as a key function of talin1 (Dedden et al., 2019). This phenomenon of opening-closing governed by ionic strength is likely mimicking the protein’s activation, i.e. opening of binding sites for partners as an interaction platform. It is worth noting that the folding and unfolding of talin1 has been thoroughly characterized by cryo-EM and SEC-MALS, and it is well-folded under the conditions tested. As our experiments suggest that talin2 also undergoes similar open-close conformational changes and is indistinguishable from talin1 in actin cosedimentation assays (Author response image 1) we conclude that our purified talin2 is functional. It is possible that the presence of talin2 in the pellet in these assays may result from physiologically relevant clustering of the protein. The propensity of talin to cluster is the subject of on-going work, and will be extensively investigated in a future publication.

**Author response image 1. sa2fig1:** Comparison of talin1 and talin2 in an actin bundling (low speed; 10,000 x g) and binding (high speed; 100,000 x g) co-sedimentation assay with pre-polymerized filamentous actin (2. 5 µM) and purified FA proteins (2.5 µM).

2) Although encapsulation of talin, vinculin and actin in GUVs is an impressive accomplishment (Figure 7), the conclusions of such a beautiful reconstitution are somewhat limited, mainly because the experiment is not sufficiently documented. The authors conclude that "these results reinforce our observations and present a direct observation of the membrane-dependent activation of talin and vinculin, leading to actin reorganization". Actually, to reach this conclusion the respective localizations of talin, vinculin and actin bundles inside the GUVs should be examined. Are talin and vinculin bound to the membrane, like in focal adhesions, or do they accumulate along the actin filament bundles? Are actin bundles floating inside GUVs without anchoring points, or bound to the inner leaflet of the membrane? These issues should be addressed through more thorough analysis of the GUV experiment.

To improve the representation of the encapsulation experiment in Figure 7, images of the individual channels (Tn, Vn, and actin) are now included in the figure, as well as a video for viewing 3D reconstructions of a representative vesicle for each condition (Figure 7—video 1). The text has been updated to better describe the observations of actin reorganization within the vesicles, including talin and vinculin localization and proximity to the membrane, in the following text:

“In both cases, any bundles present are coated with talin and vinculin, and often have at least one contact point with the membrane. […] Overall, these results reinforce our SLB experiments, and present a direct observation of the membrane-dependent activation of talin and vinculin, leading to actin reorganization.”

The actin structures and localization of talin and vinculin in these experiments do not replicate those observed at FAs, where talin and vinculin are specifically localized close to the membrane surface and do not coat the actin bundles that extend from FA protein assemblies. Though we have characterized a few of the interactions central to FA assembly, additional layers of regulation must be present to target talin and vinculin activity to discrete locations on the membrane and within FAs. Additional discussion of this has been added to the manuscript. It is likely that other FA proteins are critical for limiting interactions between activated talin, vinculin, and actin to the membrane surface, and that more elaborate reconstituted systems are necessary to better understand the complex parameters underlying FA organization. Using this experimental set-up established as a foundation, additional FA regulating proteins can be incorporated, increasing the complexity of the system, and leading to further insight into FA organization and regulation.

3) The possible role of vinculin-membrane interactions remains obscure. The authors demonstrate that removing the talin FREM domain is not sufficient to activate wild-type vinculin. Since the authors could achieve activation of wild-type vinculin only on PIP_2_-rich membranes (in the presence of talin), it would be interesting to examine, or at minimum to better discuss, the possible role of vinculin – PIP_2_ interactions, or some currently uncharacterized effects of PIP_2_ on talin conformation, in this process.

We agree that the possible role of vinculin-PIP_2_ interactions in vinculin regulation requires further investigation and is of great interest to the field. In our hands, we were unable to detect direct interactions between full-length vinculin and membrane-embedded PIP_2_. Furthermore, though we observed reduced membrane binding after mutating a reported PIP_2_-binding site in the vinculin tail (Vn^tail3Q^) (see Figure 5—figure supplement 1), this mutation had no adverse effect on talin-dependent recruitment of the full-length protein. However, we also found that full-length Vn^3Q^ showed an associated secondary effect, i.e. release of autoinhibition. This means that the disruption of the lipid binding site actually primes vinculin for interactions with talin and facilitates recruitment to the membrane via talin. As a result, we could not elucidate the potential role of vinculin-PIP_2_ interactions in vinculin recruitment to the membrane. These results, as well as a more extensive summary of past studies related to vinculin-membrane interactions and possible interpretations in a cellular context, are now covered more extensively in the Discussion (subsection “PI(4,5)P_2_ drives activation of FA proteins talin and vinculin”).

An additional explanation is that the coincidence of talin and PIP_2_ maximizes the probability of vinculin recruitment to and sustained activation at sites of focal adhesion. In this context, the release or partial release of vinculin autoinhibition would occur stochastically via “molecular breathing”, while maintaining an open, active conformation requires stabilization by additional binding partners such as talin and PIP_2_. Stable interactions between talin-vinculin and PIP_2_-vinculin might only form simultaneously; though fluctuations in vinculin head-to-tail interactions could allow either talin or PIP_2_ to bind transiently, neither interaction alone is strong enough to overcome vinculin autoinhibition for extended periods of time. However, when both talin and PIP_2_ are available to bind to vinculin, and do so at the same time, vinculin is stabilized in an open conformation, allowing actin to bind and forming a stable complex.

A final possibility is that membrane binding has a secondary effect on talin conformation, in addition to interrupting the interactions between the talin head and rod domains. The effect of membrane binding on the structure of talin and how it relates to the availability of vinculin binding sites is of particular interest, but would require extensive additional experiments. It will be the focus of future studies.